# Impacts of wildfire smoke aerosols on near-surface ozone photochemistry

Jiaqi Shen[1], Ronald C. Cohen[2,3], Glenn M. Wolfe[4], Xiaomeng Jin[1]

[1]Department of Environmental Sciences, Rutgers, The State University of New Jersey, New Brunswick, New Jersey 08901, United States
[2]Department of Chemistry, University of California Berkeley, Berkeley, California 94720, United States
[3]Department of Earth and Planetary Sciences, University of California Berkeley, Berkeley, California 94720, United States
[4]Atmospheric Chemistry and Dynamics Laboratory, NASA Goddard Space Flight Center, Greenbelt, MD 20771, United States

*Correspondence to: Xiaomeng Jin (xiaomeng.jin@rutgers.edu), ORCID: 0000-0002-6895-8464*

**Abstract.** Wildfires have been an increasing concern for the environment, yet the ozone ($O_3$) production from wildfires remains poorly characterized. Here, we aim to elucidate the role of aerosols from wildfire smoke in near-surface $O_3$ photochemistry by integrating insights from 0-D box model (F0AM) to 3-D chemical transport model (GEOS-Chem). While smoke aerosols typically inhibit $O_3$ production through heterogeneous chemical and radiative pathways, we find that for most fires, the $O_3$ enhancement driven by precursor emissions outweighs these aerosol-driven suppression effects. The relative importance of the two aerosol effects varies, with the heterogeneous chemical effect generally overshadowing the radiative effect in the far field of fires. However, near the sources of extremely large fires, the radiative effect dominates, leading to an overall suppression of $O_3$ production. By assessing the chain termination of hydrogen oxide radicals ($HO_x$) and introducing the "light-limited" regime determination in GEOS-Chem, we find that a significant portion of $O_3$ production occurred within light-limited and heterogeneous chemistry-inhibited regimes during the 2020 wildfire season in California. Building on the discovery that both aerosol and nitrogen oxide ($NO_x$) concentrations modulate aerosol influence, we demonstrate that the surface $PM_{2.5}$ to tropospheric $NO_2$ column ratio—a metric retrievable from satellite—can serve as an indicator for identifying aerosol-dominated regimes through observations.

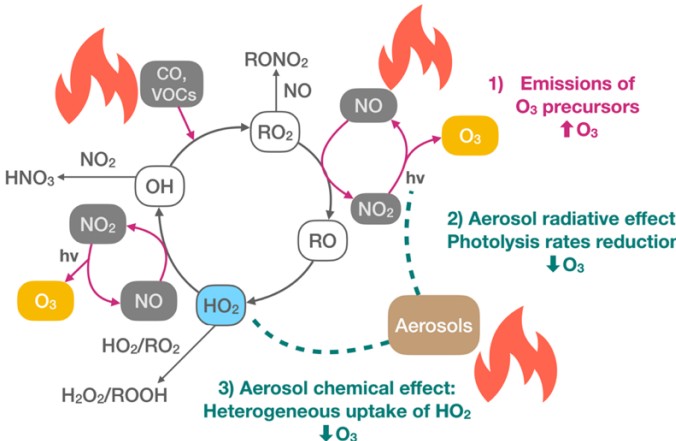

**Summary**. This study shows large chemical and radiative effects of smoke aerosols from fires on near-surface $O_3$ production. Aerosol loading and $NO_x$ levels are identified as the primary factors influencing these effects. Furthermore, we show that the surface $PM_{2.5}$ to $NO_2$ column ratio can be used as an indicator for identifying aerosol-dominated regimes, facilitating the assessments of aerosol impacts on $O_3$ formation through satellite observations.

## 1. Introduction

Over recent years, wildfires have surged in size and severity (Cattau et al., 2020; Collins et al., 2021; Hanes et al., 2019; Li and Banerjee, 2021), presenting escalating challenges to air quality, ecosystems, social economics and human health (Duane et al., 2021; Jaffe et al., 2020; Jones et al., 2022; Reid et al., 2016; Wardle et al., 2003). Wildfires release substantial amounts of carbon monoxide (CO), volatile organic compounds (VOCs), oxides of nitrogen ($NO_x$) and aerosols or particulate matter (PM) (Akagi et al., 2011). Wildfires also markedly complicate $O_3$ air pollution mitigation, as many studies have documented exceedances of the $O_3$ air quality standard and enhanced background $O_3$ level due to fires (Dreessen et al., 2016; Gong et al., 2017; Jaffe et al., 2004; Jaffe and Wigder, 2012). Fires not only emit abundant $O_3$ precursors but also provide important sources of hydrogen oxide radicals ($HO_x = OH + HO_2 +$ organic peroxy radical ($RO_2$)) through the photolysis of nitrous acid (HONO), formaldehyde (HCHO), other aldehydes and $O_3$, as well as the ozonolysis of alkenes (Jaffe and Wigder, 2012; Robinson et al., 2021; Xu et al., 2021). These radicals catalyze the chain oxidation of VOCs in the presence of $NO_x$ to produce $O_3$ (Xu et al., 2021). The $NO_x$-VOCs-radical controlled $O_3$ formation mechanism has been well-established over several decades (Pusede et al., 2014).

The impact of aerosols on $O_3$ formation, particularly in the context of wildfires, remains poorly understood. Generally, aerosol particles affect $O_3$ chemistry through two mechanisms: a radiative effect and a chemical effect. The radiative effect occurs when aerosols reduce light transmission, thereby slowing down photochemical reactions (He and Carmichael, 1999). The chemical effect refers to the role of aerosols in providing surfaces for the reactive uptake of $HO_2$, $RO_2$, oxygenated volatile organic compounds such as HCHO and reactive nitrogen species including $NO_2$, $NO_3$ and $N_2O_5$; among these chemical effects, $HO_2$ uptake dominates, especially in the daytime near-surface $O_3$

chemistry (Carlos-Cuellar et al., 2003; Ha et al., 2020; Jacob, 2000; Li et al., 2019). Aerosols typically inhibit $O_3$ formation (Benas et al., 2013; Jiang et al., 2012; Li et al., 2019; Xu et al., 2012), except in certain instances where the reduction in photolysis rates disproportionately affects $O_3$ loss more than $O_3$ production (Real et al., 2007). $O_3$ formation in wildfires exhibits considerable variability, with some studies reporting even suppressed $O_3$ in plume center or downwind areas and in Mediterranean/boreal regions (Alvarado et al., 2010; Paris et al., 2009; Strada et al., 2012; Verma et al., 2009). Model studies often invoke underestimated heterogeneous chemistry as a source of persistent bias in overpredicting $O_3$ (Jaffe and Wigder, 2012; Konovalov et al., 2012), yet, the impacts of aerosols on $O_3$ chemistry remain notably under-characterized. There is a pressing need to comprehensively evaluate the chemical and radiative effects of aerosols across different types of fires and at various stages of fire aging. Furthermore, understanding conditions under which fire emissions of $NO_x$ or VOCs or aerosols predominate is crucial for detangling the fire-related $O_3$ chemistry.

Photochemical regimes indicating $O_3$ sensitivity towards different precursor emissions have been used to guide regional air quality control strategies (Kleinman, 1994; Kleinman et al., 1997; Milford et al., 1994; Tonnesen and Dennis, 2000a, b). The two classical $O_3$ regimes are $NO_x$-limited and $NO_x$-saturated (or VOC-limited). $O_3$ production is fueled by $HO_x$ and the termination of the $HO_x$ free radical chain by either self-reaction to yield peroxides ($NO_x$-limited) or with $NO_x$ to yield $HNO_3$ and $RONO_2$ ($NO_x$-saturated) defines the regime (Ivatt et al., 2022; Sillman and He, 2002). However, large aerosol loadings—typical of wildfire smoke and many polluted areas—often complicate $O_3$ formation in ways that the classical regimes do not capture. For instance, an aerosol-inhibited regime was recently identified in heavily polluted areas of China and India, pointing to a strong impact of heterogeneous chemistry on $O_3$ formation (Ivatt et al., 2022). Moreover, dense smoke can create a dark environment that makes $O_3$ production limited by light (Jiang et al., 2012). As wildfires intensify and smoke plumes spread to downwind urban areas, understanding if and how such aerosol-inhibited behavior occurs in wildfire plumes becomes crucial for potential policy interventions and more accurate fire-related $O_3$ predictions. Therefore, in this study, we refine the current $O_3$ regime framework by introducing a new regime–the light-limited regime to better represent the role of aerosols in $O_3$ formation.

The 2020 California fires provide a valuable opportunity to study the impacts of aerosols on $O_3$ chemistry in wildfire plumes because they were especially extensive, varied in their intensity and well documented. Throughout the year, 8648 fires burned approximately 4.3 million acres across the state, with intense fire activities spanning from mid-August to November (CAL FIRE, 2020a). Fig. 1 illustrates the distribution and burned area of major fires that occurred from August to October in 2020. The widespread wildfire season in the western US in 2020, far from being an outlier, is considered a harbinger of a new norm in a warming climate (Coop et al., 2022; Xie et al., 2022). $PM_{2.5}$ pollution in western US is projected to double or even triple by the late 21[st] century under intermediate- and low-mitigation scenarios (Xie et al., 2022).

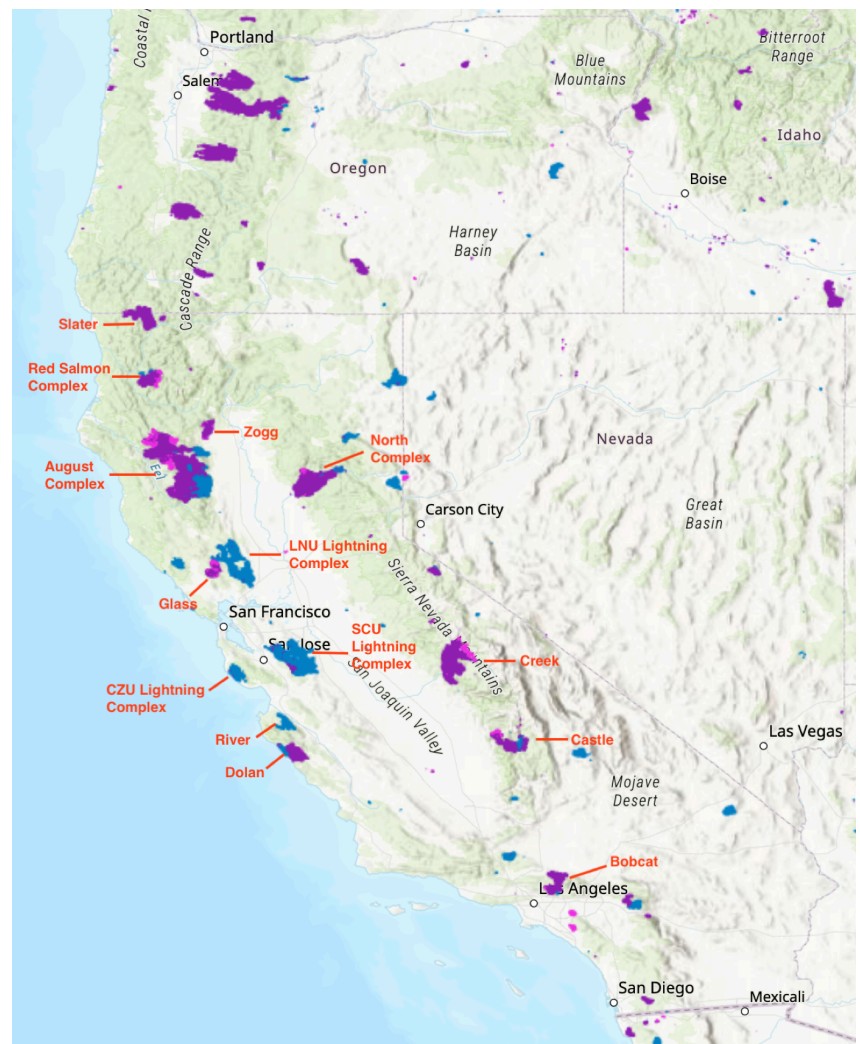

**Figure 1. Major fires during the 2020 California wildfire season (August—October). The map is sourced from NASA's Fire Information for Resource Management System (FIRMS) (NASA-FIRMS, 2025). Shaded areas represent MODIS-detected burned area, with blue, purple and pink indicating fires occurring in August, September and October, respectively.**

In this study, we employ a 3-D global chemical transport model (GEOS-Chem) and a box model (Framework for 0-D Atmospheric Modeling, F0AM) as well as observational constraints to elucidate the aerosol chemical and radiative effects on $O_3$ production in the near field and far field of fires, as well as for different types of fires. We examine the role of emissions and of aerosols in $O_3$ production and delve into the underlying processes. We provide a comprehensive evaluation of $O_3$ production regimes by introducing two additional regimes, light-limited and aerosol chemistry-inhibited, to the well-established two-regime ($NO_x$-limited and VOC-limited) classification. Furthermore, we explore the potential of the $PM_{2.5}$ to $NO_2$ ratio as an indicator for identifying aerosol-dominated regimes. We derive the threshold based on the model diagnostic approach and apply it to observation-derived $PM_{2.5}$ and $NO_2$ datasets to distinguish the aerosol-dominated $O_3$ regimes.

## 2. Materials and Methods

### 2.1 GEOS-Chem simulations

We use the GEOS-Chem (Bey et al., 2001) chemical transport model version 12.7.1 to examine the effects of aerosols on $O_3$–$NO_x$–VOCs chemistry. We run nested simulations over California regions (27° N–47° N, 110° W–130° W) with a resolution of 0.25° (latitude) × 0.3125° (longitude) and 47 vertical levels for the year 2020. The model is driven by the Goddard Earth Observation System Forward Processing product (GEOS-FP) assimilated meteorological field, with a three-hour temporal resolution for three-dimensional variables and one-hour resolution for surface variables. Boundary conditions for the simulations are generated from a global simulation at a resolution of 2° × 2.5° with one-year spin-up. The standard tropospheric chemical scheme includes detailed $O_3$–$NO_x$–VOCs–aerosol–halogen chemistry. Additionally we have incorporated the ethene and ethyne chemistry as introduced in GEOS-Chem version 13.3.0 (Kwon et al., 2021). Hourly anthropogenic emissions in the US are based on the EPA 2011 National Emission Inventory (NEI) and are scaled to 2020 using national interannual emission trends (US EPA, 2025). Fire emissions are sourced from the Global Fire Emissions Database (GFED, Version 4.1), with emissions categorized by fuel types, including tropical forest, temperate forest, boreal forest, savanna, peat and agricultural waste (Randerson et al., 2015). We allocate 65% of these fire emissions within the boundary layer (Fischer et al., 2014), so our findings primarily reflect fires that predominantly impact the boundary layer.

Photolysis rates in GEOS-Chem are calculated using the fast-JX scheme (Bian and Prather, 2002). The influence of aerosols on the photolysis rates are considered (Martin et al., 2003), with the adjustments for aerosol size distribution and optical properties in response to relative humidity changes. GEOS-Chem treats black carbon (BC) as externally mixed, making it challenging to explicitly simulate the lensing effect, where BC exhibits larger absorption when coated by a non-absorbing shell. To incorporate this effect, we apply an absorption enhancement factor (the ratio of mass absorption efficiency (MAE) with and without coating) of 1.5 to hydrophilic BC and 1 for hydrophobic BC (Wang et al., 2014).

The heterogeneous uptake of $HO_2$ is represented by a reaction probability parameterization as shown in Eq. (1), with the loss rate limited by diffusion or free molecular collision (Martin et al., 2003).

$$k = (\frac{a}{D_g} + \frac{4}{v\gamma})^{-1}A \qquad (1)$$

The first-order rate constant $k$ for the chemical loss of the gas (i.e., $HO_2$) is calculated based on the mean molecular speed ($v$), gas-phase molecular diffusion coefficient ($D_g$), aerosol radius (a), reaction probability upon impacting the aerosol surface ($\gamma$) and aerosol surface area per unit volume of air (A). Consistent with numerous modeling studies (Ivatt et al., 2022; Jacob, 2000; Li et al., 2019; Martin et al., 2003), we adopt a uniform value of 0.2 for $\gamma_{HO_2}$, aligning with the field measurements (Taketani et al., 2012; Zhou et al., 2020, 2021). GEOS-Chem assumes the same $\gamma_{HO_2}$ for all aerosol types, including organic carbon (OC), BC, sulfate-ammonium-nitrate, sea salt separated in two size bins and mineral dust in seven size bins.

130   To examine the aerosol effects on $O_3$, we conduct one BASE simulation and five perturbation simulations in GEOS-Chem, as summarized in Table 1. The difference between BASE and BASE_NO_RAD is considered as the radiative effect of all aerosols, and the difference between NO_FIRE and NO_FIRE_NO_RAD represents the radiative effect of aerosols other than fire smoke aerosols. The radiative effect of fire smoke aerosols is therefore calculated as BASE – BASE_NO_RAD – (NO_FIRE – NO_FIRE_NO_RAD). Similarly, the chemical effect of smoke aerosols is

135 calculated as BASE – BASE_NO_CHEM – (NO_FIRE – NO_FIRE_NO_CHEM). Hourly species concentrations, meteorology, photolysis rates and reaction rates for the bottom five layers of the model (approximately 0–550 m) are averaged to investigate aerosol effects on near-surface $O_3$ and perform regime calculations.

| # | Simulation Name | Description |
|---|---|---|
| (1) | BASE | |
| (2) | BASE_NO_RAD | Aerosol extinction on photolysis rates is turned off |
| (3) | BASE_NO_CHEM | Heterogeneous $HO_2$ uptake is turned off |
| (4) | NO_FIRE | Fire emissions are switched off |
| (5) | NO_FIRE_NO_RAD | Both fire emissions and aerosol radiative effect are deactivated |
| (6) | NO_FIRE_NO_CHEM | Both fire emissions and reactive uptake of $HO_2$ by aerosols are turned off |

**Table 1. Summary of the BASE simulation and five perturbation simulations conducted in GEOS-Chem.**

## 2.2 Fire plume evolution analysis

140   GEOS-Chem's Eulerian framework does not explicitly resolve individual plume pathways or their detailed evolution. We identify about 1633 fire plumes in 2020 that show clear plume patterns with an identifiable plume source and use the Hybrid Single-Particle Lagrangian Integrated Trajectory (HYSPLIT) dispersion model to calculate plume trajectories and plume age. The plume identification method is described in the work of Jin et al. (2023). Fire centers are identified using the Moderate Resolution Imaging Spectroradiometer (MODIS) Active Fire products and

145 subsequently used as starting points for calculating one-day plume dispersion using the HYSPLIT model with meteorological fields from North American Regional Reanalysis (NARR). The HYSPLIT model is run at an injection height of 1000 m and initialized at the same time of the day (18 UTC). In the absence of strong wind variability, the predicted plume trajectories should reasonably represent the progression from the near to far field of fires. The locations of the fire plumes are matched to GEOS-Chem grids to demonstrate changes in aerosol effects along the

150 plumes. In this study, we define plume age as physical age of the plume, determined as the time required for the plume to reach designated smoke-affected areas. We did not explicitly isolate fire plumes from urban influence in order to examine aerosol effects across a range of background $NO_x$ levels.

## 2.3 Box model setup

   We employ F0AM (Wolfe et al., 2016) version 4.3 to assess the effectiveness of GEOS-Chem in resolving

155 the aerosol effects on $O_3$ within fire plumes. We use the Master Chemical Mechanism (MCM) version 3.3.1 (Jenkin et al., 2015), which features a near-explicit chemical mechanism with detailed gas-phase chemical processes.

Additionally, we incorporate the heterogeneous uptake of $HO_2$ by aerosols as described in Eq. (1) and assume a monodisperse size distribution for each aerosol type.

We first evaluate whether the aerosol effects resolved in GEOS-Chem are reproducible in F0AM by initializing F0AM with output from GEOS-Chem. The fire plumes are modeled with a pseudo-Lagrangian style in F0AM, where we set the initial chemical concentrations based on GEOS-Chem grids with plume age of one hour and allow them to evolve over the subsequent five hours. Species used to initiate F0AM include CO, $O_3$, reactive nitrogen species and some VOCs (Table S2). Meteorological variables and photolysis-relevant parameters are constrained at each model step and held constant during the integration time of one hour. We adopt the F0AM's hybrid method for J-values calculations, which uses Tropospheric Ultraviolet and Visible (TUV)-calculated solar spectra but does not include explicit aerosol effects. J-values of HONO and HCHO from GEOS-Chem are applied to scale box model-calculated J-values. CO is an approximately conservative tracer (Robinson et al., 2021); we calculate the first-order dilution rate in F0AM at each model step using the temporal changes in CO concentrations along the fire plumes (Müller et al., 2016), as determined by GEOS-Chem. Configuration details of the F0AM setup are provided in Table S2. Chemical species, meteorological and photolysis variables from GEOS-Chem are matched to those in the MCM. To exhibit the aerosol effects on $O_3$, we run one base simulation and two perturbation simulations in F0AM: one eliminating the chemical effect and another removing the radiative impact of fire-related aerosols.

We further assess whether the resolution of GEOS-Chem can resolve the in-plume $O_3$ chemistry by focusing on fresh plumes in F0AM. Unlike previous setup using GEOS-Chem outputs, here we initiate F0AM with gas phase pollutants and aerosols (primarily OC and BC) for various fire types according to the GFED emission factors. We adopt aerosol effective radii of 0.035 μm for BC and 0.1 μm for OC, values that closely match GEOS-chem averages over California in 2020 at 1:30 PM local time, and assume a particle density of 1.3 g $cm^{-3}$. We convert the emission factors (g species per kg dry matter burned) to concentrations (ppb for gases and μg $m^{-3}$ for aerosols) using a fixed ratio of biomass burned per cubic meter of air. We then scale all pollutants to achieve aerosol concentrations ranging from 1 to 300 μg $m^{-3}$ at the time of emission, allowing us to explore how aerosol effects vary with fire intensity. In this approach, we set only the initial chemical and physical parameters and run the model for one hour, focusing specifically on the characteristics of fresh plumes. Photolysis rates, which we cannot directly constrain in scenarios with and without fires, are estimated based on the relationship between photolysis rate reduction and $PM_{2.5}$ mass as derived from GEOS-Chem (Figure S2). To prevent the build-up of secondary species, we set a one-day lifetime for all species by applying a first-order dilution rate of 1/86400 $s^{-1}$ and background concentrations at zero. Aerosol effects are calculated following the same method as in the F0AM–GEOS-Chem comparison.

### 2.4 Observational data

We use daily ground-based measurements of $O_3$ and $PM_{2.5}$ from the EPA Air Quality System (AQS) (EPA AQS, 2020) to evaluate the GEOS-Chem simulations. In addition, we analyse the decay of $PM_{2.5}$ and $NO_2$ within fire plumes using observationally derived datasets. Surface $PM_{2.5}$ data are from Wei et al. (2023), featuring a daily, 1 km

resolution, gapless $PM_{2.5}$ dataset spanning 2017–2022. This dataset was generated using a 4-Dimensional Space-Time Extra-Trees (4D-STET) model, which reconstructs missing satellite AOD, establishes AOD-$PM_{2.5}$ relationships and predicts high-resolution surface $PM_{2.5}$ concentrations. This observation-based 1 km product improves upon earlier 10 km datasets, providing finer spatial detail for plume analysis. Tropospheric $NO_2$ column data are sourced from
TROPOspheric Monitoring Instrument (TROPOMI) retrievals provided by Jin et al. (2023), which incorporate *a priori* profiles from GEOS-Chem simulations and explicitly account for smoke aerosols during retrieval. Both the surface $PM_{2.5}$ and tropospheric $NO_2$ column data are also used to identify $O_3$ regimes from observations (see Section 3.5).

**2.5 Photochemical regime identification**

         We determine the photochemical regimes by assessing the chain termination rates of $HO_x$ radicals, similar to
the method described in Ivatt et al. (2022). The radical termination pathways include (1) loss via $NO_x$ as indicated by the reactions $NO_2 + OH \rightarrow HNO_3$, and $RO_2 + NO \rightarrow$ alkyl nitrate ($RONO_2$), (2) $HO_x$ self-reactions, and (3) heterogenous uptake of $HO_2$ by aerosols. A predominance of $NO_x$ as the sink for $HO_x$ characterizes a $NO_x$-saturated regime. Dominance by $HO_x$ self-reactions indicates a $NO_x$-limited regime. When the rate of $HO_2$ uptake to aerosol dominates, it indicates a heterogeneous chemistry-inhibited regime. The radiative effect of aerosols, however, has not
been considered in the regime calculations. To address this issue, we account for the aerosol radiative effect on $O_3$ production by using the difference in total $HO_x$ termination rates between BASE and BASE_NO_RAD simulations ($\Delta R_{HOx}$) as a proxy. Notably, $\Delta R_{HOx}$ is not an actual chemical pathway; instead, it serves as an indicator of light availability and its influence on the photochemical activities. If $\Delta R_{HOx}$ exceeds any of the aforementioned three pathways, it suggests a light-limited regime. We use the reaction rates output from GEOS-Chem to calculate the chain
termination rates and $\Delta R_{HOx}$ in each grid box at 20:30 UTC (around 1:30 PM local time) and identify the corresponding regime based on the maximum term. We focus on 1:30 PM local time because it coincides with a period of strong solar radiation that drives ozone photochemistry and aligns with typical satellite overpass time, facilitating integration of satellite-based observations to identify chemical regimes. Monthly mean regimes are determined by averaging the magnitudes of four terms rather than counting the occurrences of each regime, to reflect the cumulative influence of
these processes over time.

         We further investigate how $PM_{2.5}$ levels influence $O_3$ photochemical regimes using GEOS-Chem. Specifically, we identify all fire-affected grid cells (those with $PM_{2.5}$ enhancement larger than 10 $\mu g\ m^{-3}$) at 20:30 UTC during 2020. For these grid cells, we calculate the $HO_x$ termination rates, determine the corresponding $O_3$ regimes, and then group the regimes by $PM_{2.5}$ concentrations to derive the probability of each regime at various $PM_{2.5}$ levels.

## 3. Results and Discussion

### 3.1 The role of smoke aerosols in $O_3$ production

We first evaluate GEOS-Chem predicted $O_3$ with daily ground measurements from the EPA AQS, as presented in Figure S1. The comparison is conducted between AQS sites and the corresponding GEOS-Chem grid cells for the year 2020 around 1 PM local time. The modeled average $O_3$ levels in California for 2020 are approximately $48 \pm 4$ ppb, in good agreement with ground observations of $44 \pm 9$ ppb ($R^2$ of 0.64).

Next, we assess the aerosol effects and the overall impact of fires on $O_3$ in GEOS-Chem in both the near and far field of fires (Figure 2). Fire pixels are categorized based on PM enhancement ($\Delta PM_{2.5}$), calculated as the difference in $PM_{2.5}$ mass between the BASE and NO_FIRE simulations for each individual grid cell. Specifically, $\Delta PM_{2.5}$ values of $<50$ $\mu g$ $m^{-3}$, $50–100$ $\mu g$ $m^{-3}$, $100–200$ $\mu g$ $m^{-3}$ and $>200$ $\mu g$ $m^{-3}$ are used to classify small, medium, large and extreme fire pixels, respectively. It reveals that for fire pixels with small to large PM enhancements, which represent the majority of fires, fires increase $O_3$ concentrations in both near and far fields, indicating the influence of fires through the emissions of substantial quantities of $O_3$ precursors outweighs the aerosol effects. Generally, fire pixels with larger PM enhancement are associated with larger increase in $O_3$ concentrations. In contrast, pixels affected by extreme fires see suppressed $O_3$ levels in their immediate vicinity, suggesting the aerosol effect overshadows the emission effect. Furthermore, this $O_3$ suppression is likely driven by the strong aerosol radiative effect associated with dense plumes near the centers of fires. In the near field of the fires, the average radiative impact on $O_3$ concentrations for extreme fire pixels is about 60 times that observed in the others. Other factors contributing to the decreased $O_3$ concentrations may be $NO_x$ titration or sequestration of $NO_x$ into peroxyacetyl nitrate (PAN) in the near field of fires (Jaffe and Wigder, 2012). For extreme fires, $O_3$ suppression by aerosols is stronger in the near field and weakens downwind, leading to a net increase in $O_3$ concentrations in the far field (Figure 2).

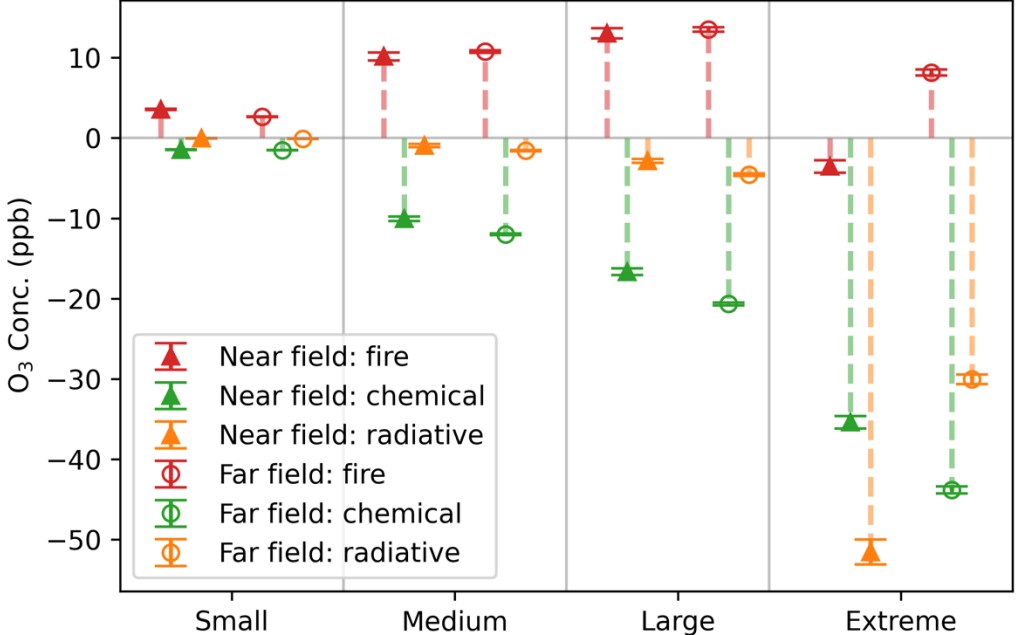

**Figure 2. Total fire effects and aerosol chemical and radiative impacts on O₃ resolved in GEOS-Chem, across near and far fields at 20:30 UTC for fire plumes in 2020. Grid cells with a plume age of 1–3 hours are marked as near field (triangles), and 4–24 hours as far field (circles). To further elucidate the dependence of aerosol impacts on PM, we classify fire pixels into different groups based on the enhancement of PM$_{2.5}$ (ΔPM$_{2.5}$) at each grid box: small (ΔPM$_{2.5}$ <50 μg m$^{-3}$), medium (50–100 μg m$^{-3}$), large (100–200 μg m$^{-3}$) and extreme (>200 μg m$^{-3}$). The total fire impact, chemical and radiative impacts on O₃ concentrations are represented by red, green and orange colors, respectively. Error bars denote standard errors. The overall fire effect is indicated by the difference in O₃ concentrations between the BASE and NO_FIRE simulations. Calculations of the aerosol effects are provided in the method section.**

Both aerosol chemical and radiative effects are shown to decrease O₃ in the fire plumes. For grid cells affected by small to large fires, the aerosol chemical effect outweighs the radiative effect. Contrary to the consistent behaviors observed in both the near-field and far-field regions for these fire pixels, those experiencing extreme PM enhancement exhibit pronounced variations. In the proximal areas of fire origins, the radiative effect on O₃ concentrations is much higher than the heterogeneous chemical effect for these extreme fire pixels. Yet, this radiative effect represents a temporary suppression of O₃ production, with its influence decaying rapidly—on average, the effect on O₃ concentrations diminishes by about half within five hours (Figure S3). Moving further from the fire centers, the chemical effect starts to dominate over the radiative effect on O₃. The aerosol impacts on O₃ concentrations, through both chemical and radiative pathways, tend to intensify as ΔPM$_{2.5}$ increases. The aerosol effects on O₃ concentrations mirror those on O₃ net production (Figure S4). However, a notable difference exists when comparing large and extreme fire pixels: while their chemical effects on O₃ production are similar in the near field (Fig. S4), extreme fires exert a stronger suppression on O₃ concentrations (Fig. 2). This discrepancy likely stems from differences in transport and

mixing. In the near field of extreme fires, $HO_x$ levels are low due to limited photochemical activity, making $HO_2$ uptake less influential on $O_3$ production. Nevertheless, extreme fires may cause greater suppression of $O_3$ concentrations near the source. As $O_3$ is transported downwind, this initial suppression can lead to a greater reduction in $O_3$ concentrations despite similar local chemical production. Additionally, extreme fires may experience slower mixing with background air, reducing dilution of ozone-suppressed air and further enhancing the decrease in $O_3$ concentrations. Overall, aerosol effects resolved in GEOS-Chem highlight the significant heterogeneous chemical influence on $O_3$ for fires and an exceptionally critical radiative effect for extreme fires.

## 3.2 Comparison between GEOS-Chem and F0AM

We first use F0AM to conduct similar experiments with GEOS-Chem output for fire plumes of different scales. We select 12 fire plumes spanning small, medium, large and extreme cases, and comparisons for each individual fire plume are shown in Figure S5. We find that the overall fire impacts on $O_3$ concentrations and the aerosol chemical and radiative effects simulated in F0AM exhibit good agreement with those resolved in GEOS-Chem across fire plumes of different scales. Although F0AM does not explicitly account for atmospheric processes such as vertical mixing, turbulent diffusion, dry and wet deposition, these factors appear to have a negligible impact (beyond their representation as dilution) on the several-hour time scale examined here. The comparison suggests that chemistry, and to a lesser extent dilution, are the leading factors explaining most variations in aerosol effects. It should be noted that although furanoid compounds markedly influence biomass burning plume chemistry under both daytime and nighttime conditions (Decker et al., 2019; Xu et al., 2021), their reactions are not represented in either the GEOS-Chem version or the MCM mechanism used in this study.

Our results indicate relatively consistent aerosol effects resolved by different numerical simulation schemes. GEOS-Chem is a global Eulerian model, which solves continuity equations on a geographically fixed frame of reference (Liu et al., 2023; Long et al., 2015), whereas in F0AM plumes are simulated in a pseudo-Lagrangian approach that follows the movement of air parcels. However, the Eulerian model struggles with an unrealistic dilution of small plumes. In our comparison, the initial chemical concentrations used in F0AM are adopted from GEOS-Chem where dilution of initial subgrid plumes has occurred. Consequently, although both GEOS-Chem and F0AM exhibit comparable results, the near-field behavior of subgrid plumes may not be accurately solved by either model.

Next, instead of initiating F0AM using GEOS-Chem simulations, we explore the aerosol influence on $O_3$ in fresh plumes by initiating F0AM with emission data from GFED. Our analysis reveals that the aerosol influence on $O_3$ depends on PM mass concentrations (Figure 3), which is consistent with findings from GEOS-Chem (Figure 2). Furthermore, at the same PM enhancement, the influence of aerosol chemical and radiative pathways on $O_3$ concentrations appears to vary distinctly among various fuel types, suggesting underlying factors beyond PM concentrations play a role in controlling aerosol influence. PM enhancement thresholds where the radiative effect outweighs the chemical effect vary by fuel type, being highest for boreal forest fires, followed by peat and temperate forest, and lowest in deforested/tropical forest, agricultural waste and savanna. In the case of temperate forest fires, even small plumes could exhibit a more pronounced aerosol radiative effect than the chemical effect in the near field.

As we control the PM magnitude, the various patterns across fuel types are due to variations in emission factors of $O_3$ precursors, particularly $NO_x$. According to GFED, emissions from the boreal forest fires exhibit the highest PM to $NO_x$ ratio, followed by those from peat, temperate forest and tropical forest fires. The lowest ratios are observed in agricultural waste and savanna burning. These results highlight that the aerosol influence on $O_3$ is not only dependent on the abundance of PM but also modulated by $NO_x$ concentrations. Higher $NO_x$ levels can suppress the chemical effect of aerosols by altering $HO_x$ loss pathways; under high-$NO_x$ conditions, more $HO_x$ is consumed by reactions with $NO_x$, leaving less $HO_x$ for heterogeneous uptake by aerosols. On the other hand, larger PM concentrations enhance $HO_x$ loss through aerosol uptake. The interplay between these two factors largely accounts for the variations in aerosol impacts on $O_3$ within fire plumes.

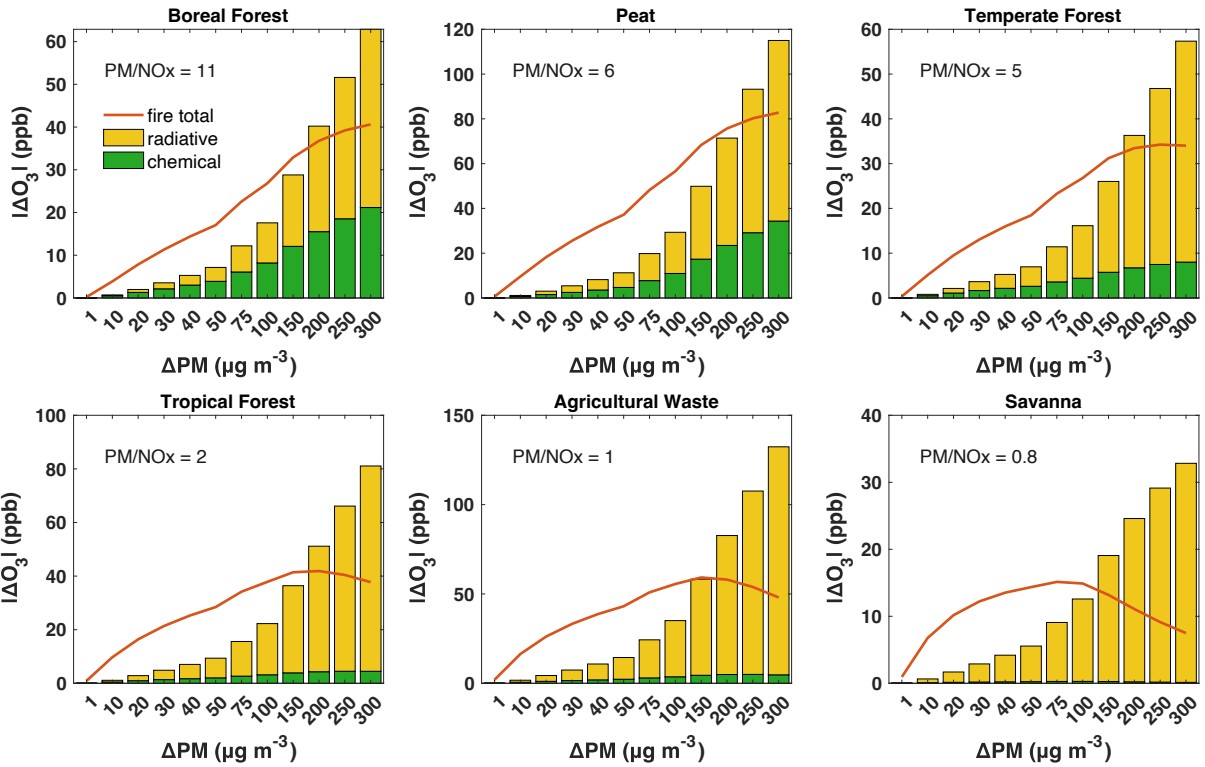

**Figure 3. The impact of aerosol chemical and radiative pathways on $O_3$ concentrations in response to intensified fires, as indicated by increasing PM enhancement, for various fuel types in the GFED emission inventory. Results are from F0AM with a one-hour run time. Orange lines denote overall $O_3$ enhancement due to fires, and green and yellow bars denote decreases in $O_3$ concentrations attributable to the aerosol heterogeneous chemical and radiative pathways. The PM to $NO_x$ emission ratio is annotated for each fuel type.**

Studies generally report dilution rates in fire plumes on the order of $10^{-5}$–$10^{-4}$ $s^{-1}$ (Decker et al., 2021; Peng et al., 2021; Rickly et al., 2022), with some studies observing rates as high as $10^{-3}$ $s^{-1}$ (Robinson et al., 2021). In Fig. 3, we adopt a dilution rate of approximately $10^{-5}$ $s^{-1}$ and we further assess its impact by increasing this rate by factors of 10 and 100 in our F0AM simulations. Under the 10× scenario (Fig. S6 (a)), the overall fire effects and aerosol effects on $O_3$ remain comparable, with similar thresholds at which the radiative effect exceeds the chemical effect. In

the 100× scenario (Fig. S6 (b)), however, these effects diminish substantially. This is likely because an e-folding timescale of 17 minutes leaves limited time for ozone production before ozone precursors and aerosols are diluted, thereby weakening the influences of fire emissions. Nevertheless, the PM enhancement threshold at which the radiative effect exceeds the chemical effect still decreases from boreal forest, peat, temperate forest, tropical forest, agricultural waste to savanna (from >300 μg m$^{-3}$ down to about 20 μg m$^{-3}$). The sensitivity test supports our findings that both PM and NO$_x$ are key factors controlling aerosol effects on O$_3$.

The dependence of aerosol effects on NO$_x$ is also evident in GEOS-Chem. Fig. S7 suggests that the radiative effect tends to surpass the chemical effect at high NO$_x$ levels. However, GEOS-Chem also indicates that the aerosol chemical effect consistently dominates the radiative effect for regular fires, a phenomenon not reproduced in F0AM. This discrepancy may arise because GEOS-Chem does not accurately resolve the aerosol effects on O$_3$ for the subgrid-scale young plumes. But for plumes that are not in the immediate vicinity of the fire source, where mixing with background air has occurred, or in the case of large-scale fires that exceed the size of a grid cell, GEOS-Chem should be capable of resolving the aerosol impacts. Additionally, for the range of PM enhancement examined here (within 300 μg m$^{-3}$), F0AM suggests that fire generally enhances O$_3$ concentrations, aligning with our findings from GEOS-Chem.

Observations of PM$_{2.5}$ and NO$_2$ within fire plumes reveal that NO$_2$ columns decay more rapidly than PM$_{2.5}$ (Fig. 4). An even steeper decline is expected for surface NO$_2$, as surface measurements are more sensitive to local emission sources compared to large-scale satellite observations (Lamsal et al., 2014). This observational finding implies that as plumes age, the aerosol heterogeneous chemical effect becomes increasingly important, as reflected by the higher PM to NO$_2$ ratio in the far field compared to near sources. This also accounts for why, in GEOS-Chem simulations, the chemical effect tends to outweigh the radiative effect away from fire origins. By integrating GEOS-Chem and box model with observational constraints, our study provides a detailed and comprehensive depiction of aerosol effects within fire plumes and the potential underlying mechanisms.

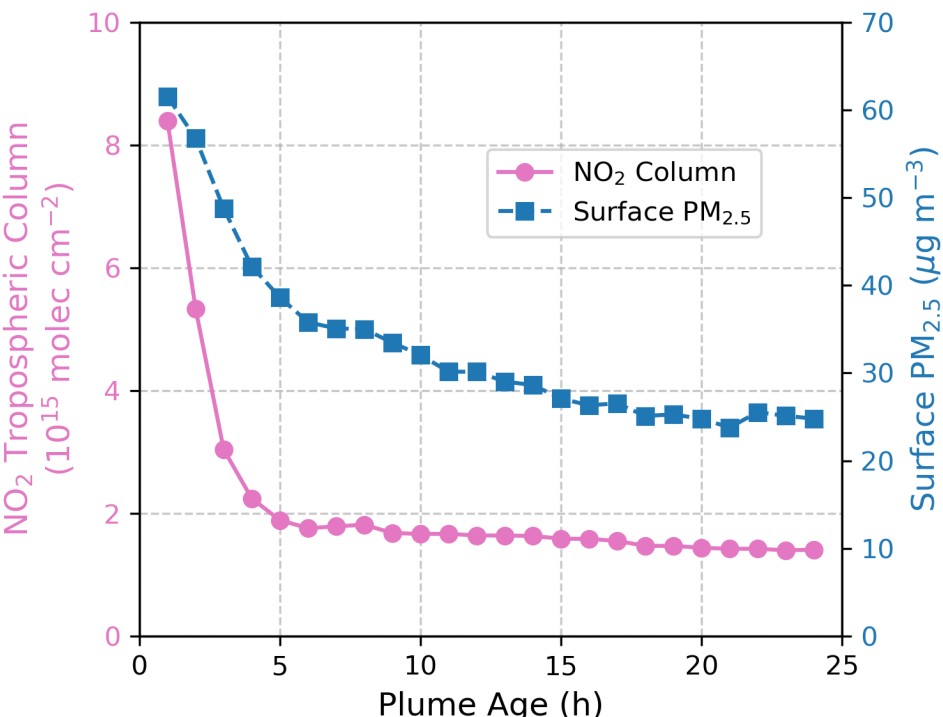

**Figure 4. Decay of NO₂ column (pink) and surface PM₂.₅ (blue) within fire plumes. Surface PM₂.₅ data are from Wei et al. (2023) and TROPOMI NO₂ tropospheric columns are from Jin et al. (2023).**

To summarize, for most fires, there is generally a net positive effect on $O_3$ concentrations. Near the source, heterogeneous chemical or radiative effects may outweigh each other depending on $NO_x$ levels. As the plumes age, $NO_x$ is rapidly consumed in the plumes, and the aerosol chemical effect tends to be increasingly important. In contrast, extremely large fires are dominated by the aerosol radiative effect, leading to an overall suppression of $O_3$ in the near field that can extend further from the fire sources. Even for these fires, the radiative effect diminishes rapidly with dilution and is eventually surpassed by the chemical effect downwind.

The importance of aerosol effects on $O_3$, especially the heterogeneous chemical effect, has been a subject of significant debate. Xu et al. (2021) found that the conceptual model based on gas phase chemistry adequately explains the $O_3$ chemistry in western US wildfire plumes ($R^2$ of 0.64) and thus aerosol heterogeneous chemical processes are likely minor. Conversely, Li et al. (2019) and Ivatt et al. (2022) highlighted a significant role of the heterogeneous chemical effect on the near-surface $O_3$ formation in eastern China and the Indo-Gangetic Plain during the mid-2010s. Even among studies that supported the importance of the aerosol chemical effect, some emphasized its significance in environments with high aerosol loadings, while others pointed to its relevance in clean suburban areas (Li et al., 2022; Xue et al., 2014).

Our findings reconcile seemingly contradictory studies by showing that the aerosol effects on $O_3$ are determined by both aerosol loading and $NO_x$ concentrations. The study by Xu et al. (2021) focused on relatively fresh plumes, which are usually associated with high $NO_x$ concentrations, where the inhibitive effects of smoke aerosols

may be secondary. However, as plumes age and both $NO_x$ and PM concentrations decay, the longer-lived accumulation mode aerosols (lifetime of 5–7 days, compared to hours to a day for $NO_x$) (Jin et al., 2021; Seinfeld and Pandis, 2016) can become more influential in $O_3$ production. The shift in the relative importance of aerosols vs. $NO_x$ may differ in urban/suburban settings, where PM and $NO_x$ can originate from different sources and possibly lead to more varied concentration patterns. $O_3$ production can be significantly impacted by heterogeneous chemistry in conditions ranging from heavily polluted areas with high aerosol loadings to cleaner areas with moderate aerosol loadings but low $NO_x$.

**3.3 Prevalence of aerosol-dominated regimes during the 2020 California fire season**

Our findings emphasize that both the heterogeneous chemical and radiative effects can significantly influence $O_3$ production depending on fire conditions. Driven by these insights, we propose a novel $O_3$ production regime, termed the "light-limited regime", which is identified through a sensitivity test in which the radiative effect is turned off and the resulting reduction in $HO_x$ availability outweighs any of the three termination pathways. Figure S8 illustrates the $O_3$ production regime over California from July to December under a no biomass burning scenario. In the absence of fire impacts, most of the areas are in $NO_x$-limited regimes during the summertime, with a $NO_x$-saturated regime in urban cores of Los Angeles and San Francisco. During the cooler months, a large number of regions shift to a VOC-limited regime.

Accounting for the impacts of fires on $O_3$ reveals significant changes in the $O_3$ production regimes during the fire season, as shown in Figure 5 (and Figure S9). Details about significant fire events and emissions during the 2020 wildfire season in California are summarized in Text S1. It is evident that numerous areas transition to either the heterogeneous chemistry-inhibited regime or the light-limited regime, which we collectively term as "aerosol-dominated regimes".

From August to October, the monthly mean proportions of grid boxes in California entering the aerosol-dominated regimes were 8.9%, 75%, and 43%, respectively (Figure 5). Specifically, 8%, 60% and 41% corresponded to the heterogeneous chemistry-inhibited regime, and 0.9%, 15%, and 1.7% were classified as light-limited regime. The impact of fires on these regimes was minimal for November, when most wildfires were contained. Furthermore, the episodic nature of wildfires caused large daily variations of the $O_3$ production regime; the heterogenous chemistry-inhibited regime had an average ± standard deviation of 19 ± 13%, 48 ± 16% and 33 ± 24% for the periods of August 16–August 31, September and October, respectively. Similarly, the light-limited regime showed 1.6 ± 1.4%, 13 ± 9.6% and 3.2 ± 5.9% for the same periods.

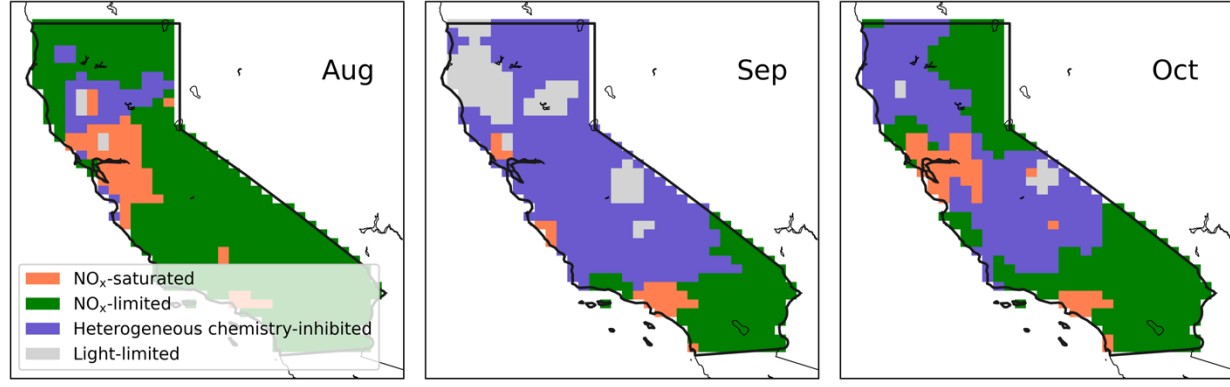

**Figure 5. Monthly-mean GEOS-Chem derived O₃ photochemical regimes at 20:30 UTC (corresponding to 13:30 local time during daylight saving and 12:30 otherwise) over California during the fire season (August to October), when fires are accounted.**

The newly defined light-limited regime extensively reflects the central areas of megafires (Figure 1). The August Complex, SCU Lightning Complex, Creek, LNU Lightning Complex, and North Complex ranked the top five fires by burned areas in 2020 (CAL FIRE, 2020b). Notably, during September, the August Complex, Creek and North Complex fires peaked, leading to extensive areas falling under the light-limited regime due to these large-scale wildfires, with the peripheral zones exhibiting heterogeneous chemistry-inhibited regime (Figure 5). The period from September 8 to 10, during the fire season, experienced the most extensive coverage of the light-limited regime across the state (32–42%, Figure S10), coinciding with significant wildfire events. Notably, despite the exceptionally large scale of the SCU and LNU Lightning Complex fires, their impacts on the light-limited regime were much less pronounced compared to the other three fires based on both daily and monthly average. A NO$_x$-saturated regime was predominant under the impact of these two wildfires. We attribute the difference in regimes to the distinct environments where fires occurred. Contrary to the fires in forest areas, the SCU and LNU fires occurred in the Bay area, an urban region characterized by significant higher background levels of NO$_x$. Elevated NO$_x$ concentrations enhance HO$_x$ termination through reactions with NO$_x$, necessitating higher aerosol concentrations for the light effect term to surpass this termination pathway. The observed reduction in the light-limited regime under high NO$_x$ conditions further corroborates our earlier findings on the interactions among aerosol effects, PM and NO$_x$.

**3.4 Uncertainties in GEOS-Chem resolved aerosol effects and O₃ regimes**

The aerosol effects and regime calculations derived from modeling are subject to uncertainties, primarily associated with the HO$_2$ uptake coefficient ($\gamma_{HO_2}$) and fire emission inventory. Due to the challenges of directly observing or constraining the aerosol heterogeneous uptake through measurements, we rely on model simulations to estimate the chemical effect. Consequently, the results are influenced by the $\gamma_{HO_2}$ values used in the analysis, a parameter that varies with aerosol types and relative humidity. A summary of $\gamma_{HO_2}$ reported in previous laboratory measurements and field studies is provided in Table S1. Organics constitute a major fraction of biomass burning aerosols. Laboratory studies measuring the uptake coefficient from single-component organics have reported values

of 0.007–0.09 for humic acid (Lakey et al., 2015), <0.01–0.13 for levoglucosan (Taketani et al., 2010) and 0.02–0.18 for dicarboxylic acids (Taketani et al., 2013), across a variety of relative humidity levels. In comparison, field studies generally report higher values (0.08–0.40) (Taketani et al., 2012; Zhou et al., 2020), likely due to the presence of copper and iron ions in the particles that are known to enhance $HO_2$ uptake (Mao et al., 2013). To our knowledge, no studies have specifically measured $\gamma_{HO_2}$ for biomass burning aerosols in field settings, but Taketani et al. (2012) reported values of 0.2–0.37 for samples strongly affected by biomass burning. To assess the impact of $\gamma_{HO_2}$ on our results, we conduct sensitivity tests using $\gamma_{HO_2}$ of 0.1 and 0.02 for a one-month simulation during September. Under the $\gamma_{HO_2} = 0.1$ scenario, aerosol effects across fire sizes are similar to Fig. 2: the aerosol chemical effect outweighs the radiative effect for small to large fire pixels, while extreme fire pixels show a pronounced radiative effect (Fig. 11 (a) (b)). Although the overall fire effect reduces $O_3$ net production rate, its influence on $O_3$ concentrations is minimal. The spatial pattern of photochemical regimes remains largely unchanged under this scenario (Fig. S12 (a)).

Given that $\gamma_{HO_2}$ measured for single-component organics likely underestimates values for ambient aerosols, the $\gamma_{HO_2} = 0.02$ case is tested as a conservative lower bond. Under this assumption, aerosol chemical and radiative effects on $O_3$ concentrations become comparable for most fire pixels, whereas extreme fire pixels continue to exhibit a pronounced radiative effect (Fig. 11 (c) (d)). Although this strong radiative effect suppresses $O_3$ production in near-field extreme fire pixels, $O_3$ concentrations still increase, possibly due to transport of ozone produced earlier near the fire source. With this substantially reduced uptake coefficient, the spatial extent of heterogeneous chemistry-inhibited regimes decreases markedly. Nevertheless, overall aerosol influences remain important, with 31% of California falling into aerosol-dominated regimes (Fig. S12 (b)). Future research measuring $\gamma_{HO_2}$ for smoke aerosols is needed to better constrain this parameter.

Furthermore, we evaluate GEOS-Chem simulations of PM$_{2.5}$ with ground-based measurements from EPA's AQS. We find that GEOS-Chem tends to overestimate PM$_{2.5}$, simulating 2020 daily average PM$_{2.5}$ levels at $24 \pm 23$ µg m$^{-3}$, compared to $12 \pm 5.5$ µg m$^{-3}$ from ground-based observations. During the fire season, modeled PM$_{2.5}$ concentrations are about 1.2, 4.1 and 2.4 times higher than the ground observations in August, September and October, respectively. Outside the peak fire months, the agreement improves, with modeled PM$_{2.5}$ concentrations being 0.6, 1.4 and 0.9 times the observed values in July, November and December, respectively. The overestimates of PM$_{2.5}$ is likely driven by overestimated fire emissions in GFED (Qiu et al., 2024). These comparisons, however, are limited by factors such as the sparse ground observations (~72 sites for PM$_{2.5}$), the potential unrepresentativeness of a single site for the coarse grid in GEOS-Chem, and the GEOS-Chem modeled decay of PM further from the fires. To assess the potential impacts of model overestimates on our analysis, we perform additional simulations by scaling monthly biomass burning emissions based on the model–observation comparisons. Specifically, GFED fire emissions are adjusted by dividing total emissions by 0.6, 1.2, 4.1, 2.4, 1.4 and 0.9 for July through December, respectively. Despite the substantial reduction in overall fire emissions, the aerosol and total fire effects on $O_3$ concentrations for most fires remain consistent with Fig. 2, whereas the radiative effect for extreme fires declines markedly in both the near and far field due to reduced aerosol loading (Fig. S11 (e) (f)). Aerosol-dominated regimes still accounted for about 7%, 54%

and 17% of the total area in August, September and October, respectively (Fig. S13). Notably, aerosol-dominated regimes remain dominant in September during the 2020 fire season.

**3.5 What is the PM$_{2.5}$ threshold for reaching aerosol-dominated regimes?**

Recognizing that the regime classification discussed above may be affected by model inputs and performance, we further explore how these model-based findings can be applied to observational data, with a primary focus on identifying aerosol-dominated regimes. We first investigate whether PM$_{2.5}$ as an indicator of aerosol concentrations can be used to identify the regime shift. Fig. 6 (a) shows the average fractional contribution of each HO$_x$ termination pathway at various PM$_{2.5}$ levels. As PM levels increase, HO$_x$ loss via self-reaction declines, while aerosol heterogeneous uptake and photolysis reduction effects become increasingly dominant. Fig. 6 (b) exhibits the probability of each regime at various PM$_{2.5}$ levels. Low PM$_{2.5}$ levels are usually associated with a NO$_x$-limited regime. The heterogeneous chemistry-inhibited regime is more likely to occur as PM$_{2.5}$ levels increase until the light-limited regime overshadows it at extremely high PM$_{2.5}$ concentrations. At PM$_{2.5}$ concentration of 30 μg m$^{-3}$, O$_3$ production already transitions to the heterogeneous chemistry-inhibited regime in most areas under the impact of fires. A considerably higher PM$_{2.5}$ concentration (~500 μg m$^{-3}$) is required to enter the light-limited regime. We observe a similar pattern of HO$_x$ losses and regime shifts when reducing the $\gamma_{HO_2}$ value to 0.1, as shown in Figure S14. In this calculation, the PM$_{2.5}$ threshold for shifting to a heterogeneous chemistry-inhibited regime increases slightly from 30 to 40 μg m$^{-3}$.

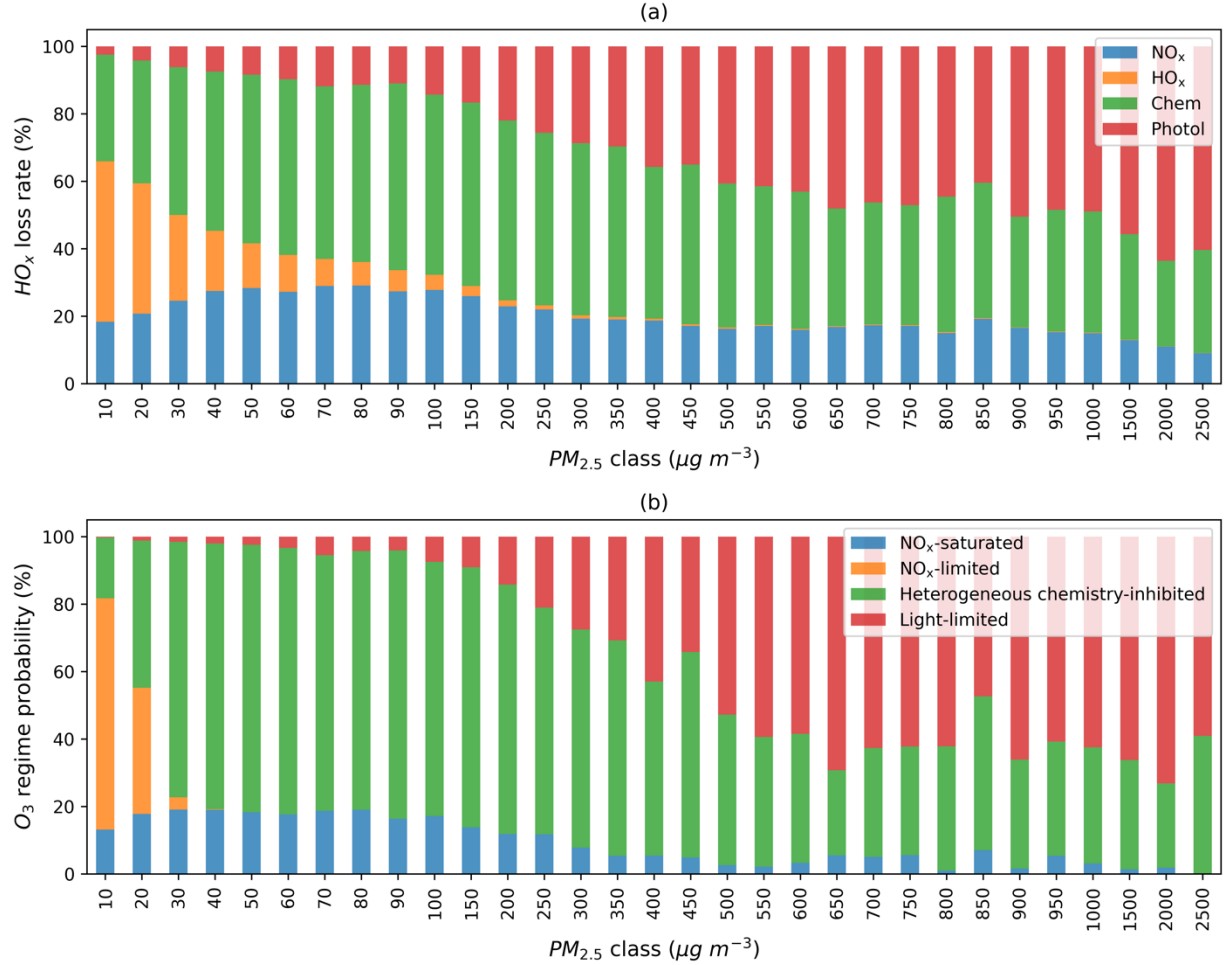

**Figure 6. (a) Average fractional contribution of the four HOₓ termination terms to the total. (b) Probability distribution of grid boxes across different photochemical regimes at various PM₂.₅ levels. The analysis includes all fire-affected grid boxes at 20:30 UTC on all days in 2020, identified based on ΔPM₂.₅ > 10 µg m⁻³. PM₂.₅ classes denote rounded total PM₂.₅ concentrations.**

It is important to note that classifying a regime as "heterogeneous chemistry-inhibited" or "light-limited" does not necessarily imply a net suppression of $O_3$. The regime classification approach based on $HO_x$ termination rate does not directly compare with the aerosol and emission effects quantified in Section 3.1 and 3.2. For example, a "heterogeneous chemistry-inhibited" regime indicates that $HO_2$ uptake is the largest sink of $HO_x$, but does not imply that the combined aerosol chemical and radiative effects outweigh the influence of VOC and $NO_x$ emissions. As shown in Fig. 2, large fires with PM₂.₅ enhancement of 200 µg m⁻³ still exhibit net $O_3$ increases despite strong heterogeneous chemical effects.

Further investigations uncover that the PM₂.₅ threshold required for most grid boxes to transition to a heterogeneous chemistry-inhibited regime is highly dependent on $NO_x$ concentrations (Figure 7 (a)). Here we categorize $NO_x$ concentrations into four classes: 0–1, 1–10, 10–25 and >25 ppb, and the PM₂.₅ thresholds likely to

485 induce aerosol-dominated regimes are approximately 18, 25, 185 and 320 $\mu g\ m^{-3}$, correspondingly. We primarily focus on fire plumes in this study, but grid boxes not affected by fires appear to exhibit similar trends in the probability of aerosol-dominated regimes. These results support our earlier findings that in scenarios with high $NO_x$ concentrations, more PM is needed to attain a comparable level of aerosol contribution as observed in low $NO_x$ scenarios. As $NO_x$ concentrations increase, $HO_x$ levels tend to decrease (Fig. 7 (b)), which necessitates higher PM levels for aerosol

effects to surpass the emission effects. Since $PM_{2.5}$ and $NO_2$ can be derived from ground-based or satellite observations, we explore how their ratio can be used to imply aerosol-dominated regimes. While a surface $PM_{2.5}/NO_2$ ratio may seem more straightforward based on our analysis, the limited spatial coverage of surface $NO_2$ measurements poses a challenge. Tropospheric $NO_2$ column data, which are closely related to surface sources and have been widely used in $O_3$ sensitivity analyses (Martin et al., 2004), offer a practical alternative. When combined with high resolution and

gapless surface $PM_{2.5}$ estimates derived from the integration of observations and machine learning, the $PM_{2.5}/NO_2$ column ratio serves as a proxy to constrain aerosol effects on near-surface $O_3$ production. Indeed, we find a clear relationship between this ratio and the likelihood of aerosol-dominated regimes (Figure 7 (c)). When the ratio ($PM_{2.5}/NO_2$ column) reaches about 20 $(\mu g\ m^{-3})/(10^{15}$ molecules $cm^{-2})$, the aerosol-dominated regimes are likely to prevail, and will consistently be dominant at higher ratios.

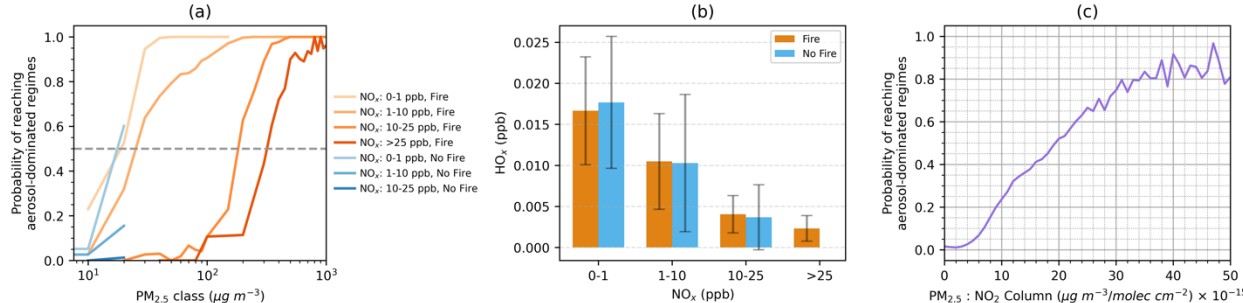

**Figure 7. (a) Probability of achieving aerosol-dominated regimes in response to varying $PM_{2.5}$ and $NO_x$ concentrations, differentiated by fire-impacted (orange) and non-fire (blue) grid boxes. The dashed line marks the thresholds where half of the grid boxes enter aerosol-dominated regimes. (b) $HO_x$ concentrations across different $NO_x$ concentration bins. (c) Relationship between the surface $PM_{2.5}$ to $NO_2$ column ratio and the**
505 **probability of reaching aerosol-dominated regimes, combining both fire-impacted and non-fire grid boxes.**

We therefore adopt a threshold of 20 $(\mu g\ m^{-3})/(10^{15}$ molecules $cm^{-2})$ for $PM_{2.5}/NO_2$ identifying aerosol-dominated regimes and apply it to satellite-derived surface $PM_{2.5}$ and tropospheric $NO_2$ column data. The resulting aerosol-dominated regimes are highlighted in red in Fig. 8. Overall, the observation-based aerosol-dominated regimes in Fig. 8 align well with the model-based classification in Fig. 5. Both methods reveal similar spatial distributions:

aerosol-dominated regimes were widespread in September, peaked in the northern and central region in October, but exhibited relatively larger discrepancies in August. Spatially, the observation-based method estimates that approximately 20%, 47% and 16% of the state fell within aerosol-dominated regimes from August to October; these values are larger than the model estimates for August but somewhat smaller for September. Despite these differences, the general agreement between the two regime classifications underscores the significant role of aerosols in surface

$O_3$ photochemistry under wildfire conditions. This analysis also highlights the utility of satellite-derived $PM_{2.5}$ to $NO_2$

ratio for pinpointing aerosol-dominated regimes. Applying this metric to other regions and environments may need further investigation. While the comparison of fire and urban plumes is beyond the scope of this study, it is worth noting that fire and urban plumes may differ substantially in emissions and aerosol composition and thus the $O_3$ chemistry. Future research is therefore warranted to incorporate more sophisticated representations of these differences. Additionally, it may be valuable to compare the robustness of this metric with a fully satellite-based indicator, such as the $AOD/NO_2$ ratio.

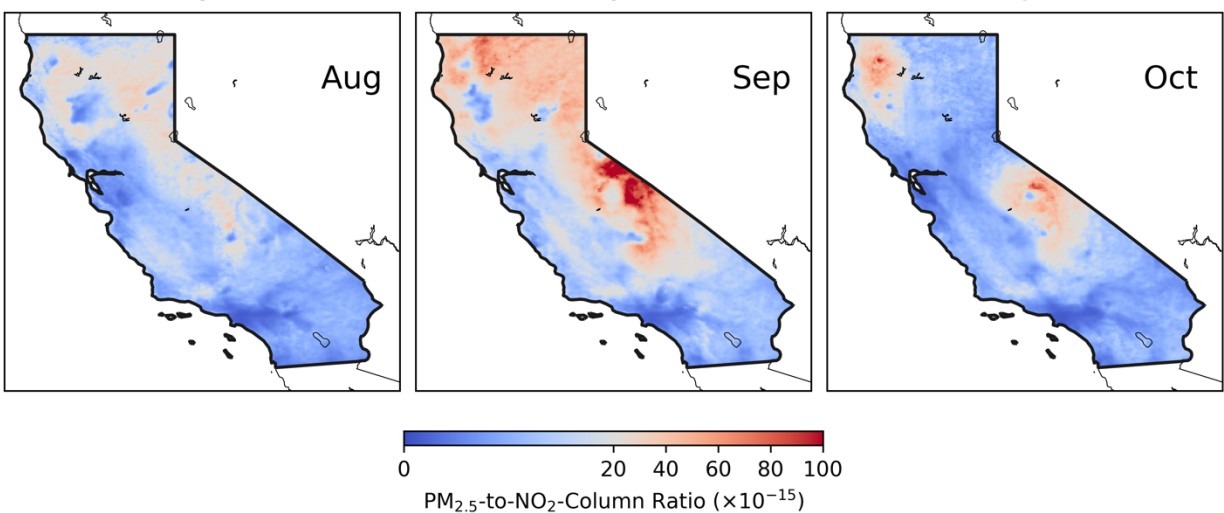

**Figure 8. Monthly mean $O_3$ photochemical regimes identified using the surface $PM_{2.5}$ to TROPOMI $NO_2$ column ratio over California from August to October. Red colors represent aerosol-dominated regimes, while blue colors indicate $NO_x$-limited or $NO_x$-saturated regimes. Monthly mean $PM_{2.5}$ and $NO_2$ are used to calculate the ratio, with a threshold of 20 (μg m$^{-3}$)/(10$^{15}$ molecules cm$^{-2}$) applied to identify aerosol-dominated regimes.**

**4. Conclusion**

Aerosols typically suppress surface $O_3$ formation through heterogeneous uptake of $HO_2$ and the reduction of photolysis rates, yet both pathways are difficult to observe or measure directly. Here, we combine GEOS-Chem, F0AM box model and observational constraints to examine aerosol effects on $O_3$ formation. We found that for most fires, $O_3$ concentrations increase because emissions of $O_3$ precursors outweigh aerosol effects. In contrast, during extreme large fires, the strong radiative effect may lead to an $O_3$ suppression near the fire sources. As plumes age, the aerosol chemical effect becomes more pronounced. To represent these effects, we introduce the aerosol heterogenous chemistry-inhibited and light-limited regimes into GEOS-Chem. Our results suggest that aerosol-dominated regimes played a significant role during the 2020 wildfire season in California.

Aerosol loading and $NO_x$ levels are the key factors governing aerosol effects on near-surface $O_3$ formation. Under $NO_x$-saturated and aerosol-dominated regimes, $O_3$ chemistry becomes $HO_x$-limited. Higher $NO_x$ reduces aerosol effects by driving more $HO_x$ to react with $NO_x$. These results imply that even at similar aerosol concentrations, fire and urban plumes are likely to experience different levels of aerosol effects and fall in distinct photochemical

regimes. Within wildfires, areas are apt to achieve the heterogeneous chemistry-inhibited regime when $PM_{2.5}$ concentrations approach tens of $\mu g\ m^{-3}$. However, the typically high $NO_x$ concentrations in urban areas may preclude the emergence of aerosol-dominated regimes in these regions. These insights have significant implications for $O_3$ pollution in downwind urban areas. Previous studies have pointed out that VOC-rich wildfire plumes can enhance $O_3$ pollution when they mix into high-$NO_x$ urban plumes (Jin et al., 2023; Xu et al., 2021). This study, however, unveils an additional, hidden downside of urban high $NO_x$: it obscures aerosol effects that would otherwise help reduce $O_3$, thereby exacerbating $O_3$ pollution relative to scenarios where wildfire smoke penetrates rural or suburban areas. It suggests that reducing $NO_x$ concentrations in urban downwind areas could yield further benefits for mitigating $O_3$ pollution under fire conditions.

In addition to the diagnostic modeling approach for identifying aerosol-dominated regimes, we propose using the surface $PM_{2.5}$ to $NO_2$ column ratio as an indicator. When combined with the widely used HCHO to $NO_2$ ratio (FNR) for identifying $NO_x$-limited or $NO_x$-saturated regimes with satellite remote sensing (Itahashi et al., 2022; Jin et al., 2020; Souri et al., 2020), this enables a comprehensive identification of $O_3$ regimes on a global scale using observation-based $NO_2$, HCHO and $PM_{2.5}$. However, challenges remain for identifying $O_3$ regimes under wildfire conditions due to retrieval uncertainties in thick smoke plumes and significant primary HCHO emissions that may compromise its effectiveness as an indicator of VOC reactivity (Liao et al., 2021). More work is needed to evaluate the reliability of FNR thresholds in wildfire plumes and to refine $PM_{2.5}$ to $NO_2$ thresholds under diverse environmental settings to improve our ability to characterize photochemical regimes.

**Data Availability**

The data and scripts for the regime classification in this study are openly available at https://github.com/Jiaqi-Shen/Shen_et_al_fire_chemistry_manuscript.

**Competing interests**

The contact author has declared that none of the authors has any competing interests.

**Author Contributions**

J.S.: Methodology, Formal Analysis, Data Curation, Visualization, Writing - Original Draft. R.C.C.: Conceptualization, Funding Acquisition, Writing - Review & Editing. G.M.W.: Methodology, Writing - Review & Editing. X.J.: Conceptualization, Methodology, Funding Acquisition, Writing - Review & Editing. All authors have given approval to the final version of the manuscript.

**Acknowledgements**

This study was supported by NOAA Climate Program Office's Atmospheric Chemistry, Carbon Cycle, and Climate program (grant number: NA22OAR4310199) and NASA Aura Science Team and Atmospheric Composition Aura Science Team and Atmospheric Composition Modeling and Analysis Program (grant number: 80NSSC23K1004). This research used the computational cluster resource provided by the Office of Advanced Research Computing (OARC) at Rutgers, The State University of New Jersey. We thank Jean Rivera-Rios from Rutgers University for insightful discussions on organic nitrate chemistry, Ke Li from Nanjing University of Information Science and Technology for discussions on GEOS-Chem simulations and Jing Wei from University of Maryland for providing data support on surface PM$_{2.5}$. GMW acknowledges support from the NASA Tropospheric Composition program.

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
