# Peer review of "Impacts of wildfire smoke aerosols on near-surface ozone photochemistry"

_EGUsphere, 2025_

## Author Comment (AC1)

We thank the reviewers for their constructive feedback that has helped us to improve the manuscript. Our responses to their comments are provided below, with reviewer comments in black, our responses in blue.

**Reviewer 1**

**General Comments**

This study investigates the impact of aerosols from wildfires on near-surface ozone production and suppression. It uses a variety of tools – a global model, box model, surface data, and satellite data – to perform a comprehensive analysis of the aerosol-driven processes that impact ozone levels in fires of various sizes. The authors have considered many scientific, physical, and model-driven processes that could affect their results and have done a good job explaining their reasoning. I think this paper will be acceptable for publication after minor revisions.

**Specific Comments**

- The paper would benefit by adding a figure to the Introduction that shows the extent of the fires over California in 2020. The authors could consider figures that show ground heat, smoke extent, etc., averaged over Sep, Aug-Oct, periods of high fire activity, etc. This figure would be useful to cite when introducing the 2020 fires on line 75 and again in line 349 after "reflects the central areas of megafires". In the paragraph beginning on line 75, you should also reference Text S1.

Thank you for the great suggestion. We have added a figure showing the spatial distribution and MODIS burned area of the primary fires from August to October in 2020.

This new figure is now referenced in both the Introduction, where we describe the 2020 California wildfire season, and in Section 3.3, where we discuss the distribution of light-limited regimes.

The revised text in the Introduction now reads: "Throughout the year, 8648 fires burned approximately 4.3 million acres across the state, with intense fire activities spanning from mid-August to November (CAL FIRE, 2020a). **Fig. 1 illustrates the distribution and burned area of major fires that occurred from August to October in 2020.**"

In Section 3.3, we now state: "The newly defined light-limited regime extensively reflects the central areas of megafires **(Figure 1)**."

The remaining figures in the main text have been relabeled accordingly following the addition of the new Figure 1.

[Figure]

**Figure 1. Major fires during the 2020 California wildfire season (August—October). The map is sourced from NASA's Fire Information for Resource Management System (FIRMS) (NASA-FIRMS, 2025). Shaded areas represent MODIS-detected burned area, with blue, purple and pink indicating fires occurring in August, September and October, respectively.**

- Table S1 is introduced on line 121 but it feels out-of-place (or unnecessary) at this point in the manuscript. I understood its inclusion once I got to Section 3.4 and you performed a sensitivity test on. I recommend stating earlier in the paper that you will be testing the impact of so that Table S1 makes more sense. Alternatively, you could remove Table S1 since you list a range of values in lines 369-376. If you keep Table S1, you can just reference it in lines 369-376 instead of reiterating the values in the main text.

Thank you. We have removed the sentence "A summary of $\gamma_{HO_2}$ reported in previous laboratory measurements and field studies is provided in Table S1." from line 121 and instead referred to Table S1 in the Uncertainties section. We believe it is helpful to list several $\gamma_{HO_2}$ values in the main text that may be relevant to fire aerosols considered in this study. The sentence "**A summary of $\gamma_{HO_2}$ reported in previous laboratory measurements and field studies is provided in Table S1.**" now follows the statement

"Consequently, the results are influenced by the $\gamma_{HO_2}$ values used in the analysis, a parameter that varies with aerosol types and relative humidity." in Section 3.4.

- Your method begins all HYSPLIT simulations at 18 UTC (line 147) and all figures are shown at 13:30 LT. Can you explain that further?

We focus on 13:30 local time for ozone photochemistry because (1) it coincides with a period of strong solar radiation when ozone photochemistry is active, and (2) it aligns with the typical overpass time of satellite instruments, allowing us to leverage satellite observational data to identify chemical regimes.

The HYSPLIT data analysis is used to identify smoke plumes, and we follow the method in Jin et al. (2023) and use the MODIS Active Fire products to determine fire locations. While this dataset provides daily fire center information, it does not specify the exact ignition time. Because our primary goal with HYSPLIT is to roughly distinguish the near field and far field and given that large-scale wind patterns usually do not change drastically over several hours, the precise starting time is less critical. Furthermore, the 2020 California wildfire season was characterized by many large and long-lasting fires, which further reduces the significance of the exact hour. Therefore, we assume fires start a few hours before being detected by satellite (i.e. at 18 UTC, corresponding to 11 am in the Daylight Saving Time) and initiate our HYSPLIT simulations at that time.

- Consider commenting on the impact of using a monodisperse size distribution for all aerosol types in F0AM (line 155) compared to 2 or 7 size bins in GEOS-Chem (line 122).

Thanks for the thoughtful comment. Given the limitation of F0AM in resolving aerosol-phase chemistry and size distributions, we adopt a monodisperse aerosol size treatment as a simplified and practical assumption. To evaluate the performance of F0AM in simulating aerosol effects, we conducted a comparison between GEOS-Chem and F0AM, where F0AM model is set based on the monodisperse assumption and F0AM model is initialized based on GEOS-Chem. The comparison encompasses small, medium, large and extreme large fires (Figure S5). The results show that F0AM closely reproduces the GEOS-Chem simulations, suggesting that the use of a monodisperse size distribution introduces minimal impact on our simulations of aerosol effects in F0AM.

- The cutoff between near-field and far-field fire effects is 3 hours (Figure 1). How did you choose this cutoff value? Similarly, in lines 233-234 you state that the radiative effect diminishes by about half within 5 hours. Can you provide support (e.g., a figure in the SI) to support this?

We use a 3-hour cutoff between near-field and far-field effects based on multiple references (Peng et al., 2021; Robinson et al., 2021; Xu et al., 2021). While there is no strict universal definition, studies commonly characterize the first several hours of smoke evaluation as near-field.

Regarding our statement that the radiative effect diminishes by about half within five hours, we now provide supporting evidence in the supplement (Figure S3). The figure shows that the radiative effect for extreme fire pixels decays to approximately a half within five hours.

[Figure]

**Figure S3.** Aerosol effects on $O_3$ concentrations as a function of plume age for fire pixels grouped by size.

- Line 231: Avoid using the word "significant" if you aren't referring to a statistical analysis.

We have changed the word "significant" to "pronounced". The revised text reads: "Contrary to the consistent behaviors observed in both the near-field and far-field regions for these fire pixels, those experiencing extreme PM enhancement exhibit **pronounced** variations." We have also corrected other similar usage in the manuscript.

- The authors have shown that the $PM_5 : NO_2$ column ratio could be an indicator of aerosol-dominated regimes (Section 3.5). The authors have clearly explained their methods and results regarding this topic, but I suggest reconsidering defining a specific threshold value (20 (ug m$^{-3}$)/($10^{15}$ molecules cm$^{-2}$)). Previous studies seem to show that the $HCHO : NO_2$ ratio is better used as a qualitative indicator of $NO_x$ and VOC influences on a region, rather than a quantitative indicator, and I think that this $PM_{2.5} : NO_2$ ratio may be similar. You studied this indicator over a specific region, time, and fire(s) which is useful, but may not be applicable in other regions and times. You mention these considerations in the Conclusion, but I recommend changing the discussion at the end of Section 3.5 to focus on why this ratio is generally useful rather than stating why a specific value of the ratio is useful. Additionally, please comment on why a ratio between a surface quantity ($PM_{2.5}$) and a column quantity ($NO_2$) is or isn't useful.

Thank you for your thoughtful comments. We have added justification for why using a combined surface-column metric to identify aerosol-dominated regimes, before we introduce the $PM_{2.5}/NO_2$ column ratio. We have also revised the discussion at the end of Section 3.5 to shift the focus away from a specific threshold and toward the general utility of the ratio and directions for future work.

"Since $PM_{2.5}$ and $NO_2$ can be derived from ground-based or satellite observations, we explore how their ratio can be used to imply aerosol-dominated regimes. **While a surface $PM_{2.5}/NO_2$ ratio may seem more straightforward based on our analysis, the limited spatial coverage of surface $NO_2$ measurements poses a challenge. Tropospheric $NO_2$ column data, which are closely related to surface sources and have been widely used in $O_3$ sensitivity analyses (Martin et al., 2004), offer a practical alternative. When combined with high resolution and gapless surface $PM_{2.5}$ estimates derived from the integration of observations and machine learning, the $PM_{2.5}/NO_2$ column ratio serves as a proxy to constrain aerosol effects on near-surface $O_3$ production. Indeed, we find a clear relationship between this ratio and the likelihood of aerosol-dominated regimes (Figure 7 (c)).** When the ratio ($PM_{2.5}/NO_2$ column) reaches about 20 ($\mu g\ m^{-3}$)/($10^{15}$ molecules $cm^{-2}$), the aerosol-dominated regimes are likely to prevail, and will consistently be dominant at higher ratios."

"This analysis also highlights the utility of satellite-derived $PM_{2.5}$ to $NO_2$ ratio for pinpointing aerosol-dominated regimes. **Applying this metric to other regions and environments may need further investigation. While the comparison of fire and urban plumes is beyond the scope of this study, it is worth noting that fire and urban plumes may differ substantially in emissions and aerosol composition and thus the $O_3$ chemistry. Future research is therefore warranted to incorporate more sophisticated representations of these differences. Additionally, it may be valuable to compare the robustness of this metric with a fully satellite-based indicator, such as the $AOD/NO_2$ ratio.**"

**Technical Corrections**

- Line 16: The terms "positive" and "negative" are somewhat confusing here, since positive could mean positive ozone formation potential, or it could mean positive human impact by reducing ozone. I recommend modifying these terms to "ozone-producing" and "ozone-mitigating", or something similar.

Thank you for the helpful suggestion. We have revised the text as:

"While smoke aerosols typically inhibit $O_3$ production through heterogeneous chemical and radiative pathways, we find that for most fires, **the $O_3$ enhancement driven by precursor emissions outweighs these aerosol-driven suppression effects**."

- Line 61: Remove comma between "…precursor emissions" and "have been used…"

Removed, thank you.

- Line 85: Rephrase "delve into the reasons underlying the processes" to "delve into the underlying processes".

Fixed, thank you.

- Line 101: add "the": "anthropogenic emissions in the US are represented…"

We have added "the".

- Line 102: You need references for the NEI data and when saying that the NEI data was "scaled based on the national interannual variation in emissions".

We have added the reference and revised the sentence: "**Hourly anthropogenic emissions in the US are based on the EPA 2011 National Emission Inventory (NEI) and are scaled to 2020 using national interannual emission trends (US EPA, 2025).**"

- The paragraph beginning on line 124 would be better if the information was put into a table.

Thanks for the helpful suggestion. We have put this information into a table:

**"To examine the aerosol effects on O₃, we conduct one BASE simulation and five perturbation simulations in GEOS-Chem, as summarized in Table 1."**

| # | Simulation Name | Description |
|---|---|---|
| (1) | BASE | |
| (2) | BASE_NO_RAD | Aerosol extinction on photolysis rates is turned off |
| (3) | BASE_NO_CHEM | Heterogeneous $HO_2$ uptake is turned off |
| (4) | NO_FIRE | Fire emissions are switched off |
| (5) | NO_FIRE_NO_RAD | Both fire emissions and aerosol radiative effect are deactivated |
| (6) | NO_FIRE_NO_CHEM | Both fire emissions and reactive uptake of $HO_2$ by aerosols are turned off |

**Table 1. Summary of the BASE simulation and five perturbation simulations conducted in GEOS-Chem.**

- Line 136: Remove "SI" before "Figure S1". There are some other locations in the text where this should also be done.

Okay, we have removed "SI" before "Figure S1" and other relevant locations.

- Lines 136-138: This information should be somewhere in the Results section, not the Methods section. Also, the AQS data should be described/cited in Section 2.4.

We have cited the EPA AQS data in Section 2.4: "**We use daily ground-based measurements of O₃ and PM₂.₅ from the EPA Air Quality System (AQS) (EPA AQS, 2020) to evaluate the GEOS-Chem simulations.**"

In addition, we have moved the model evaluation results to the beginning of Results and Discussion (Section 3.1): "**We first evaluate GEOS-Chem predicted O₃ with daily ground measurements from the EPA AQS, as presented in Figure S1. The comparison is conducted between AQS sites and the corresponding GEOS-Chem grid cells for the year 2020 around 1 PM local time. The modeled average O₃ levels in California for 2020 are approximately 48 ± 4 ppb, in good agreement with ground observations of 44 ± 9 ppb ($R^2$ of 0.64).**

Next, we assess…"

- Line 152: Spell out the acronym for F0AM the first time you use it.

We spelled out the acronym F0AM as "Framework for 0-D Atmospheric Modeling" upon its first mention in the final paragraph of the Introduction: "In this study, we employ a 3-D global chemical transport model (GEOS-Chem) and a box model (Framework for 0-D Atmospheric Modeling, F0AM) as well as observational constraints to elucidate the aerosol chemical and radiative effects on O₃ production in the near field and far field of fires, as well as for different types of fires."

- Line 153: Add "the" before "Master Chemical Mechanism".

Thank you, we have added "the".

- Line 161: Cite Table S2 after saying "…and some VOCs".

We have referenced Table S2: "Species used to initiate F0AM include CO, $O_3$, reactive nitrogen species and some VOCs **(Table S2)**."

- Table S2 is somewhat confusing; I think that what it's telling the reader is what GEOS-Chem output variables are used as inputs to various F0AM runs. I recommend renaming the column labels: "F0AM inputs" ◊ "GEOS-Chem output variables used as inputs to F0AM"; "F0AM-base" ◊ "GEOS-Chem simulation corresponding to F0AM base case"; etc.

Thanks for your suggestion. We have revised the column label "F0AM inputs" to "GEOS-Chem output variables used as inputs to F0AM" in Table S2 for improved clarity. As Table S2 describes the setup of F0AM simulations, we have retained the names F0AM-base, F0AM-no_rad, and F0AM-no_chem, as we believe they are concise and effectively convey the corresponding simulation scenarios.

- Line 163: define TUV.

We have defined TUV and the revised text reads "We adopt the F0AM's hybrid method for J-values calculations, which uses **Tropospheric Ultraviolet and Visible (TUV)**-calculated solar spectra but does not include explicit aerosol effects."

- Lines 183-185: Consider adding more detail about the $PM_5$ surface data from Wei et al. (2023).

We have added more details regarding how the $PM_{2.5}$ surface data were generated. The revised paragraph now reads:

"In addition, we analyse the decay of $PM_{2.5}$ and $NO_2$ within fire plumes using observationally derived datasets. **Surface $PM_{2.5}$ data are from Wei et al. (2023), featuring a daily, 1 km resolution, gapless $PM_{2.5}$ dataset spanning 2017–2022. This dataset was generated using a 4-Dimensional Space-Time Extra-Trees (4D-STET) model, which reconstructs missing satellite AOD, establishes AOD-$PM_{2.5}$ relationships and predicts high-resolution surface $PM_{2.5}$ concentrations. This observation-based 1 km product improves upon earlier 10 km datasets, providing finer spatial detail for plume analysis.**"

- Line 188: Consider adding "(see Section 2.5)" at the end of this sentence.

Thank you for the suggestion. We have added a reference to the relevant section in the Results and Discussion where we use this observation-based ratio to infer $O_3$ regimes. The revised sentence now reads: "Both the surface $PM_{2.5}$ and tropospheric $NO_2$ column data are also used to identify $O_3$ regimes from observations **(see Section 3.5)**."

- Line 198: Consider changing "" to "" since it represents a difference in reaction rates rather than a difference in concentrations.

We have changed "$\Delta HO_x$" to "$\Delta R_{HOx}$" in all relevant places to better represent difference in $HO_x$ reaction rates.

- Line 215: Should "$NO_x$ titration, sequestration of $NO_x$" be "$NO_x$ titration or sequestration of $NO_x$"?

Okay, we have revised the text as "Other factors contributing to the decreased $O_3$ concentrations may be $NO_x$ titration **or** sequestration of $NO_x$ into peroxyacetyl nitrate (PAN) in the near field of fires (Jaffe and Wigder, 2012)."

- Lines 216-217: Is this sentence contradictory? If ozone suppression extends to distant areas of extreme fires, why would that lead to an ozone increase?

Great point. We have revised the text as "**For extreme fires, $O_3$ suppression by aerosols is stronger in the near field and weakens downwind, leading to a net increase in $O_3$ concentrations in the far field (Figure 2).**"

- Line 260: At the end of the sentence, reference Figure 1: "…consistent with findings from GEOS-Chem (Figure 1)".

We have referenced Figure 2 (formerly Figure 1) in the sentence: "Our analysis reveals that the aerosol influence on $O_3$ depends on PM mass concentrations (Figure 3), which is consistent with findings from GEOS-Chem **(Figure 2)**."

- Lines 264-265: Should "highest" and "lowest" be swapped?

Thank you for your comment. We have carefully reviewed the sentence and the supporting analysis. The statement "PM enhancement thresholds where the radiative effect outweighs the chemical effect vary by fuel type, being highest for boreal forest fires, followed by peat and temperate forest, and lowest in deforested/tropical forest, agricultural waste and savanna." is consistent with our findings. For example, in savanna fires, the aerosol radiative effect becomes dominant even with minor PM enhancement, whereas for boreal forest fires, it only becomes dominant at much higher PM enhancement ($\sim 75$ μg m$^{-3}$). Therefore, the threshold of PM enhancement where the aerosol radiative effect outweighs the chemical effect is larger for boreal forest compared to savanna, and the ordering of "highest" and "lowest" in the sentence is correct and does not require modification.

- Line 267: Rephrase: "$NO_x$ in particular" ◊ "particularly $NO_x$".

We have revised this to be "As we control the PM magnitude, the various patterns across fuel types are due to variations in emission factors of $O_3$ precursors, **particularly $NO_x$.**"

- Line 282: Add "that": "Figure 1 indicates that for PM".

We have added "that".

- Line 290: Why is a steeper decline expected for surface $NO_x$ compared to column $NO_x$? Do you have data or a reference to support this?

Thank you for your question. Difference between satellite column $NO_2$ and ground-based surface $NO_2$ measurement are primarily due to spatial inhomogeneity in $NO_2$ distributions. Lamsal et al. (2014) pointed out that $NO_2$ in the lower troposphere is short-lived and concentrated near emission sources, whereas satellite retrievals represent an average over a broader area, which can smooth out spatial gradients. Additionally, Knepp et al. (2015) observed that changes in column $NO_2$ can lag behind those at the surface. These findings suggest that surface $NO_2$ is more sensitive to local emission sources. Therefore, we expect a steeper decline in surface $NO_2$ compared to column $NO_2$ downwind of emission sources, such as fires. We have added more context to this sentence, and it now reads:

"Observations of $PM_{2.5}$ and $NO_2$ within fire plumes reveal that $NO_2$ columns decay more rapidly than $PM_{2.5}$ (Fig. 3). An even steeper decline is expected for surface $NO_2$, **as surface measurements are**

**more sensitive to local emission sources compared to large-scale satellite observations (Lamsal et al., 2014)**."

- Why are Figure 4 and Figure S6 split into 2 separate figures? If you keep them separate, modify the Figure 4 caption to list the correct months.

Thank you. We kept the August, September and October plot in the main text while the remaining months in the supplement because we primarily focus on the impact of aerosols on $O_3$ production during the fire seasons. We have corrected the Figure 4 caption to be "Figure 4. Monthly-mean GEOS-Chem derived $O_3$ photochemical regimes at 20:30 UTC (corresponding to 13:30 local time during daylight saving and 12:30 otherwise) over California **during the fire season (August to October)**, when fires are accounted".

- Lines 369-376: Listing these studies and data values is repetitive with Table S1.

Thank you for pointing this out. We now reference Table S1 in this section rather than repeating the information in the Methods. Specifically, we added the sentence: "**A summary of $\gamma_{HO_2}$ reported in previous laboratory measurements and field studies is provided in Table S1.**" before Lines 369–376. The text that follows serves as a description of the table content, with an emphasis on values relevant to fire aerosols. We believe this change reduces repetition and clarifies the motivation for our sensitivity test.

- Lines 380-382: Reference Figure S1.

Thank you but Figure S1 has been referenced earlier, here we focus on the evaluation of GEOS-Chem modeled $PM_{2.5}$ rather than $O_3$.

- Lines 387-389: This sentence ("To assess the potential…model-observation comparisons") requires more description, either in the main text or in the SI.

We included additional details in the main text on how fire emissions were scaled for the additional regime calculations. The revised paragraph now reads:

"During the fire season, modeled $PM_{2.5}$ concentrations are about 1.2, 4.1 and 2.4 times higher than the ground observations in August, September and October, respectively. **Outside the peak fire months, the agreement improves, with modeled $PM_{2.5}$ concentrations being 0.6, 1.4 and 0.9 times the observed values in July, November and December, respectively.** …To assess the potential impacts of model overestimates on our analysis, we perform additional simulations by scaling monthly biomass burning emissions based on the model–observation comparisons. **Specifically, GFED fire emissions are adjusted by dividing total emissions by 0.6, 1.2, 4.1, 2.4, 1.4 and 0.9 for July through December, respectively.**"

- Line 399: How is the "probability of each regime at various $PM_5$ levels" calculated? This should be described in the Methods section.

Thanks for the suggestion, we have added a description of this analysis at the end of Section 2.5 in the Methods. The new text reads as follows:

"**We further investigate how $PM_{2.5}$ levels influence $O_3$ photochemical regimes using GEOS-Chem. Specifically, we identify all fire-affected grid cells (those with $PM_{2.5}$ enhancement larger than 10 μg m$^{-3}$) at 20:30 UTC during 2020. For these grid cells, we calculate the $HO_x$ termination rates, determine the corresponding $O_3$ regimes, and then group the regimes by $PM_{2.5}$ concentrations to derive the probability of each regime at various $PM_{2.5}$ levels.**"

- Line 434: Change "calcification" to "classification".

Thanks for catching the typo, we have corrected it.

- Line 443: Add "and": "differ substantially in emissions and aerosol composition".

Thank you, we have added "and" to the text.

**Reviewer 2**

Summary:

This study examines how wildfire smoke aerosols influence near-surface ozone production during 2020 California fire season, using a combination of 3D chemical transport model (GEOS-Chem), 0D box model (F0AM), as well as surface and satellite observations. The authors quantify aerosol impacts through two pathways: (a) heterogeneous uptake of $HO_x$ radicals, which suppresses ozone formation, and (b) radiative effects that reduce photolysis rates. They further assess how the relative importance of these pathways varies by location (near-field vs. far-field), chemical environment (low-NOx vs. high-NOx), and biomass fuel type. Finally, the study introduces the conceptual definitions of "heterogeneous chemistry-inhibited" and "light-limited" regimes and proposes observational thresholds to diagnose these regimes.

Major Comments:

The manuscript addresses a timely and important scientific question, and it is clear that the authors have invested considerable effort and thought into this work. While many aspects are well-executed and meets publication quality, I have substantive concerns for certain parts that need to be addressed before publication, and thus recommend major revisions.

First, the authors employ the box model (F0AM) in two different configurations. In the first setup, concentrations are initialized by GEOS-Chem, and key inputs (meteorology, boundary conditions, photolysis rates, and dilution rate) are all constrained by GEOS-Chem at each time step. As expected, the F0AM results largely mirror GEOS-Chem outputs, and I don't feel it added any new insight or validation. If the authors have their reasoning to retain this set of results, I'd suggest assessing the uncertainties around dilution rate (which is not explicitly stated now) more thoroughly. If the dilution rate is set large, F0AM might be basically reproducing GEOS-Chem-derived boundary condition values; a sensitivity analysis to dilution rate will be necessary for any F0AM-derived conclusions.

Thank you for your thoughtful comments and suggestions. While we expect F0AM to reproduce GEOS-Chem outputs with initiated with the chemical species, meteorological conditions and dilution rates from GEOS-Chem, we believe this comparison remains informative for a few reasons. (1) They are different models: GEOS-Chem is Eulerian model, whereas F0AM is applied in a pseudo-Lagrangian framework. (2) F0AM is a simplified model with chemistry and dilution, other processes such as vertical mixing, diffusion and deposition in GEOS-Chem are not represented in F0AM. (3) Unlike the log-normal size distribution in GEOS-Chem, aerosol radii in F0AM are simplified by assuming a monodisperse distribution for each aerosol type. Given these differences, we believe the comparison is useful for validating consistency and ensuring that the divergence in subsequent F0AM results from GEOS-Chem is not due to model structural differences.

In response to your excellent suggestion, we have now included a sensitivity analysis on the dilution rate by increasing it by factors of 10 and 100. The F0AM sensitivity tests were conducted for aerosol effects across different fuel types. We did not apply this test to the F0AM and GEOS-Chem comparison, as our intention in that setup was for F0AM to reproduce GEOS-Chem output using consistent inputs. The sensitivity test results, now presented in Section 3.2 and Figure S6, show that:

**"Studies generally report dilution rates in fire plumes on the order of $10^{-5}$–$10^{-4}$ s$^{-1}$ (Decker et al., 2021; Peng et al., 2021; Rickly et al., 2022), with some studies observing rates as high as $10^{-3}$ s$^{-1}$ (Robinson et al., 2021). In Fig. 3, we adopt a dilution rate of approximately $10^{-5}$ s$^{-1}$ and we further assess its impact by increasing this rate by factors of 10 and 100 in our F0AM simulations. Under the 10× scenario (Fig. S6 (a)), the overall fire effects and aerosol effects on $O_3$ remain comparable, with similar thresholds at which the radiative effect exceeds the chemical effect. In the 100× scenario (Fig. S6 (b)), however, these effects diminish substantially. This is likely because an e-folding timescale of 17 minutes leaves limited time for ozone production before ozone precursors and aerosols are diluted, thereby weakening the influences of fire emissions. Nevertheless, the PM enhancement threshold at which the radiative effect exceeds the chemical effect still decreases from boreal forest, peat, temperate forest, tropical forest, agricultural waste to savanna (from >300 µg m$^{-3}$ down to about 20 µg m$^{-3}$). The sensitivity test supports our findings that both PM and NO$_x$ are key factors controlling aerosol effects on $O_3$."**

(a)

[Figure]

[Figure]

(b)

[Figure]

**Figure S6. Sensitivity tests of aerosol effects across various fuel types in F0AM, with dilution rates increased by (a) 10× and (b) 100× relative to the original dilution rate of 1/86400~1×10⁻⁵ s⁻¹.**

In the F0AM second setup, it is driven by GFED emissions without concentration constraints. This raises greater concern, as box models driven solely by emissions, particularly in the absence of observation constraints, can produce unrealistically high or low concentrations due to large inherent uncertainties. The manuscript does not provide any form of performance evaluation for this setup, yet several important conclusions (e.g., the effects of biomass fuel type or $NO_x$ levels on ozone chemistry) are derived from it. A performance evaluation against observations is needed to establish the credibility of these results.

The F0AM second setup is run for an hour to simulate fresh fire plumes. On such short timescales, other processes like mixing and deposition, beyond their representation as dilution, are expected to have limited impact. With emissions, chemistry and an observationally constrained dilution rate, we believe F0AM reasonably simulate the near-field $O_3$ chemistry. However, we agree that it is valuable to validate F0AM results against aircraft or ground-based measurements. While some studies have used F0AM to simulate specific wildfire plumes, these are typically stand-alone analyses focused on individual events. Further, the limited availability of field data spanning diverse fuel types, emission magnitudes and $NO_x$ conditions make it challenging to develop a comprehensive understanding of $O_3$ chemistry in fire plumes.

Our goal here is to complement the GEOS-Chem analysis using a box model to isolate and highlight the underlying factors driving variability in aerosol effects. We acknowledge that neither GEOS-Chem or F0AM fully represents real-world complexity, and thus we combine insights from both models to better understand fire chemistry. The idealized F0AM simulations suggest that aerosol effects on $O_3$ are modulated by $NO_x$ levels—an important finding that is also supported by GEOS-Chem simulations.

Furthermore, we believe the F0AM analysis is useful as it helps identify critical factors such as fuel type in the $O_3$ chemistry that can guide future research directions.

To further strengthen the conclusion that aerosol effects depend on $NO_x$, which is supported by both F0AM and GEOS-Chem, we have added the following to the manuscript:

"The dependence of aerosol effects on $NO_x$ is also evident in GEOS-Chem. **Fig. S7 suggests that the radiative effect tends to surpass the chemical effect at high $NO_x$ levels.**"

[Figure]

**Figure S7.** Dependence of aerosol chemical and radiative effects on $NO_x$ concentrations at 20:30 UTC based on GEOS-Chem simulations. The analysis includes all fire pixels within identified plumes.

Second, the uptake coefficient for $HO_x$ radicals, which is central to the heterogeneous chemistry pathway, is assumed to be 0.2, near the upper end of reported values ($10^{-3}$ to $10^{-1}$ magnitudes). Although a sensitivity test is presented at 0.1, this remains within the same order of magnitude and does not capture the full span of possible values. Since many conclusions rely on this parameter, a broader sensitivity analysis including lower uptake coefficients is recommended, along with clarification on which conclusions remain robust under more conservative assumptions.

Thank you for this thoughtful suggestion. Regarding the $HO_2$ uptake coefficient, modeling studies have commonly used $\gamma_{HO_2}$ of 0.2 (Bloss et al., 2005; Ivatt et al., 2022; Li et al., 2019; Zhang et al., 2022). As noted in the manuscript (Section 3.4), while some values in Table S1 appear low, measurements of these single-component organics likely underestimate uptake coefficient for ambient aerosols, as they do not account for the influence of trace metals. That said, we agree that conducting a broader sensitivity analysis is beneficial.

In response, we have conducted additional GEOS-Chem simulations using $\gamma_{HO_2}$ of 0.02 to indicate a conservative lower bound. We have also extended the sensitivity analysis for $\gamma_{HO_2}$ of 0.1 and for the reduced fire emissions scenarios to include not only regime calculations but also aerosol effects, as

suggested by reviewer 3. Section 3.4 has been renamed as "**3.4 Uncertainties in GEOS-Chem resolved aerosol effects and O₃ regimes**," and the revised text now reads:

"... **Under the $\gamma_{HO_2}$= 0.1 scenario, aerosol effects across fire sizes are similar to Fig. 2: the aerosol chemical effect outweighs the radiative effect for small to large fire pixels, while extreme fire pixels show a pronounced radiative effect (Fig. 11 (a) (b)). Although the overall fire effect reduces O₃ net production rate, its influence on O₃ concentrations is minimal.** The spatial pattern of photochemical regimes remains largely unchanged under this scenario (Fig. S12 (a)).

**Given that $\gamma_{HO_2}$ measured for single-component organics likely underestimates values for ambient aerosols, the $\gamma_{HO_2}$ = 0.02 case is tested as a conservative lower bond. Under this assumption, aerosol chemical and radiative effects on O₃ concentrations become comparable for most fire pixels, whereas extreme fire pixels continue to exhibit a pronounced radiative effect (Fig. 11 (c) (d)). Although this strong radiative effect suppresses O₃ production in near-field extreme fire pixels, O₃ concentrations still increase, possibly due to transport of ozone produced earlier near the fire source. With this substantially reduced uptake coefficient, the spatial extent of heterogeneous chemistry-inhibited regimes decreases markedly. Nevertheless, overall aerosol influences remain important, with 31% of California falling into aerosol-dominated regimes (Fig. S12 (b)).** Future research measuring $\gamma_{HO_2}$ for smoke aerosols is needed to better constrain this parameter.

Furthermore, we evaluate GEOS-Chem simulations of PM₂.₅ with ground-based measurements from EPA's AQS. ... **Specifically, GFED fire emissions are adjusted by dividing total emissions by 0.6, 1.2, 4.1, 2.4, 1.4 and 0.9 for July through December, respectively. Despite the substantial reduction in overall fire emissions, the aerosol and total fire effects on O₃ concentrations for most fires remain consistent with Fig. 2, whereas the radiative effect for extreme fires declines markedly in both the near and far field due to reduced aerosol loading (Fig. S11 (e) (f)).** Aerosol-dominated regimes still accounted for about 7%, 54% and 17% of the total area in August, September and October, respectively (Fig. S13). Notably, aerosol-dominated regimes remain dominant in September during the 2020 fire season."

(a)                               (b)

[Figure]

(c)                               (d)

[Figure]

**Figure S11**. Sensitivity tests of total fire effects and aerosol effects on $O_3$ concentrations **(a, c, e)** and $O_3$ net production rate **(b, d, f)** at 20:30 UTC for fire pixels of different sizes in both near and far field. Panels **(a) and (b)** use $\gamma_{HO_2}$ of 0.1 and include all 470 plumes from September 2020. Panels **(c) and (d)** use a lower $\gamma_{HO_2}$ value of 0.02, based on all September plumes. Panels **(e) and (f)** are based on adjusted GFED emissions and include 1347 plumes identified between July and December 2020.

[Figure]

**Figure S12.** Monthly-mean O$_3$ photochemical regimes at 20:30 UTC over California for September 2020, under the BASE scenario with **(a)** $\gamma_{HO_2}$ = 0.1 and **(b)** $\gamma_{HO_2}$ = 0.02.

Minor Comments

Line 102: NEI is annual - how was the temporal allocation done? What data sources in particular were used for interannual scaling?

GEOS-Chem has hourly and monthly version of NEI 2011 and we used hourly resolution in our simulations. We used the air pollutant emission trends data provided by EPA for interannual scaling. We have incorporated this information and added the reference. The revised text reads as follows:

**"Hourly anthropogenic emissions in the US are based on the EPA 2011 National Emission Inventory (NEI) and are scaled to 2020 using national interannual emission trends (US EPA, 2025)."**

Line 201: Please clarify the choice of 1:30 PM local time.

We have added a sentence to clarify the rationale for selecting 1:30 PM local time after it is first mentioned:

**"We focus on 1:30 PM local time because it coincides with a period of strong solar radiation that drives ozone photochemistry and aligns with typical satellite overpass time, facilitating integration of satellite-based observations to identify chemical regimes."**

Line 205: Consider adding a brief summary of fire distribution (spatial & temporal) in the main text for context.

Thank you for the helpful suggestion. We have added a map in the Introduction showing the major fires during the 2020 California wildfire season, with different colors indicating the months in which the fires occurred.

[Figure]

**Figure 1. Major fires during the 2020 California wildfire season (August—October). The map is sourced from NASA's Fire Information for Resource Management System (FIRMS) (NASA-FIRMS, 2025). Shaded areas represent MODIS-detected burned area, with blue, purple and pink indicating fires occurring in August, September and October, respectively.**

Line 208: Suggest changing "positively influence O₃" to "increase O₃" for clarity.

Changed, thank you.

Line 211: How many fires fall into the "extreme" category? Is there a threshold where aerosol effects outweigh emissions effects?

Following the subsequent comment, we have updated the aerosol effects figure to include fire plumes from the entire year of 2020, rather than limiting the analysis to September. This analysis is based on fire pixels, with approximately 4370 classified as extreme fire pixels. We did not explicitly determine a precise threshold at which aerosol effects outweigh the emission effects. This is because our analysis is intended to qualitatively illustrate how aerosol effects and emission effects evolve within fire plumes, rather than to

establish a definitive turning point. The classification into different fire intensity categories is approximate and represent average behaviors across a large number of fires.

Figure 1: You mentioned GEOS-Chem is run for the whole 2020 - why does figure 1 only include September 2020 results?

Thank you for pointing this out. We have extended our analysis to include fire plumes from the entire year of 2020. We have updated figures in both the main text and the supplement accordingly, and revised the relevant text to reflect this change. The aerosol effects and overall conclusions remain consistent with those based on the original analysis for September 2020.

Section 2.2: "We identify about **1633 fire plumes in 2020** that show clear plume patterns with an identifiable plume source and use the Hybrid Single-Particle Lagrangian Integrated Trajectory (HYSPLIT) dispersion model to calculate plume trajectories and plume age. The plume identification method is described in the work of Jin et al. (2023)."

[Figure]

**Figure 2.** Total fire effects and aerosol chemical and radiative impacts on $O_3$ resolved in GEOS-Chem, across near and far fields at 20:30 UTC for fire plumes **in 2020**. Grid cells with a plume age of 1–3 hours are marked as near field (triangles), and 4–24 hours as far field (circles). To further elucidate the dependence of aerosol impacts on PM, we classify fire pixels into different groups based on the enhancement of $PM_{2.5}$ ($\Delta PM_{2.5}$) at each grid box: small ($\Delta PM_{2.5} <50$ μg m$^{-3}$), medium (50–100 μg m$^{-3}$), large (100–200 μg m$^{-3}$) and extreme (>200 μg m$^{-3}$). The total fire impact, chemical and radiative impacts on $O_3$ concentrations are represented by red, green and orange colors, respectively. Error bars denote standard errors. The overall fire effect is indicated by the difference in $O_3$ concentrations between the BASE and NO_FIRE simulations. Calculations of the aerosol effects are provided in the method section.

[Figure]

**Figure S4**. The overall fire impact and the two aerosol effects on $O_3$ net production rate at 20:30 UTC for **all fire pixels in California during 2020**, classified by PM enhancement and proximity to fire centers. The colors red, green and orange depict the total fire impact, aerosol chemical effect and radiative effect, respectively. Triangles and circles denote the near and far field, respectively, with error bars indicating standard errors.

Line 220: I found this term "plume age" somehow confusing; based on your lines 149-150 definition, my understanding is that it's grid-specific and reflects the transport time (and thus distance) from fire source, not the post-fire evolution time, right?

Yes. The plume age definition in our study represents physical plume age. The fire plume selection method follows Jin et al. (2023), where fires are identified using MODIS Active Fire products. However, exact ignition time of each fire is often unknown. Additionally, there are many large and long-lasting fires during the California fire season, making it hard to track the exact fire evolution time. Therefore, our analysis primarily focuses on capturing the near-field and far-field behaviors of fire plumes based on their physical distance from the source. To reduce the confusion, we have revised the sentence to: "**In this study, we define plume age as physical age of the plume, determined as the time required for the plume to reach designated smoke-affected areas.**"

Line 222: is PM enhanced defined based on daily averages?

PM enhancement refers to the increase in PM concentrations at each grid cell due to fires at 20:30 UTC, rather than daily averages.

Line 300: I thought it's the emission effects dominate here, right? Maybe you mean "near the source, heterogeneous chemical may or may not outweigh radiative effects depending on NOx levels"?

Thanks for pointing out the confusion. We have corrected the sentence to be: "Near the source, **heterogeneous chemical or radiative effects may outweigh each other** depending on $NO_x$ levels.".

Line 345: Are values averaged over full month or fire days only?

The values are averaged over the full month rather than fire days only.

Line 395: Please confirm whether results are from GEOS-Chem.

Yes, to improve clarity, we have included a more detailed explanation of how these results were obtained at the end of Section 2.5 in the Methods. The new text reads as follows:

**"We further investigate how $PM_{2.5}$ levels influence $O_3$ photochemical regimes using GEOS-Chem. Specifically, we identify all fire-affected grid cells (those with $PM_{2.5}$ enhancement larger than 10 $\mu g \ m^{-3}$) at 20:30 UTC during 2020. For these grid cells, we calculate the $HO_x$ termination rates, determine the corresponding $O_3$ regimes, and then group the regimes by $PM_{2.5}$ concentrations to derive the probability of each regime at various $PM_{2.5}$ levels."**

Line 401: You note heterogeneous update effects dominate at PM2.5 >30 µg/m³, but Fig 1 shows net $O_3$ increases up to 200 µg/m³. Please reconcile and add more explanation here.

We appreciate your careful consideration. The difference stems from how we define and compare aerosol and emission effects in different sections. In Section 3.1 and 3.2, we directly quantify the two aerosol effects and compare them with emission effects. In contrast, Section 3.3 to 3.5 discusses photochemical regimes based on $HO_x$ termination pathways, including heterogeneous chemistry-inhibited, light-limited, $NO_x$-limited, and $NO_x$-saturated regimes. Here, "dominance" simply means that among these four radical-terminating pathways, one is the largest—it does not necessarily imply that aerosol effects outweigh the ozone-producing emission effects or not.

In Fig. 6 (formerly Fig. 5) we show that as PM concentrations reach about 30 $\mu g \ m^{-3}$, the heterogeneous chemistry effect dominates over the other three pathways and areas are most likely to fall in heterogeneous chemistry-inhibited regimes. However, this does not guarantee a net $O_3$ reduction if the emission impacts are still stronger overall. Indeed, in Fig. 2 we see that for large fires with PM enhancement within 200 µg/m³, there is still an average increase in $O_3$, meaning that emission-driven ozone production remains dominant. Therefore, while regime classification identifies the largest $HO_x$ sink, it does not by itself indicate net $O_3$ enhancement or suppression. These two perspectives—one quantifying net $O_3$ changes, the other identifying the primary $HO_x$ sink—are not contradictory but rather address different metrics.

To clarify this point and reduce confusion, we have added a paragraph in Section 3.5 following the discussion of Fig. 6. The new paragraph reads:

**"It is important to note that classifying a regime as "heterogeneous chemistry-inhibited" or "light-limited" does not necessarily imply a net suppression of $O_3$. The regime classification approach based on $HO_x$ termination rate does not directly compare with the aerosol and emission effects quantified in Section 3.1 and 3.2. For example, a "heterogeneous chemistry-inhibited" regime indicates that $HO_2$ uptake is the largest sink of $HO_x$, but does not imply that the combined aerosol chemical and radiative effects outweigh the influence of VOC and $NO_x$ emissions. As shown in Fig. 2, large fires with $PM_{2.5}$ enhancement of 200 $\mu g \ m^{-3}$ still exhibit net $O_3$ increases despite strong heterogeneous chemical effects."**

Line 404: Replace "significant" unless referring to statistical significance.

Thanks, we have replaced "significantly" to "considerably" and revised similar instances throughout the manuscript.

Line 410: Are results shown for fire days or all of 2020?

This O₃ regime analysis was performed using fire-affected grid cells, and we included all such grid cells at 20:30 UTC for all days in 2020. Fire pixels were selected based on $\Delta PM_{2.5}$ greater than 10 µg m⁻³. To make it clearer, we have revised the figure caption as follows:

"Figure 6. (a) Average fractional contribution of the four HOₓ termination terms to the total. (b) Probability distribution of grid boxes across different photochemical regimes at various $PM_{2.5}$ levels. The analysis includes **all fire-affected grid boxes at 20:30 UTC on all days in 2020, identified based on $\Delta PM_{2.5} > 10$ µg m⁻³**. $PM_{2.5}$ classes denote rounded total $PM_{2.5}$ concentrations."

Figure 6: Consider repositioning the color legend in panel (a).

Thanks for the suggestion. We have moved the color legend to the right side of panel (a) to improve readability. Additionally, as suggested by reviewer 3, we have added a panel (b) to exhibit HOₓ levels among different NOₓ size bins. Please see the updated figure below.

[Figure]

**Figure 7.** (a) Probability of achieving aerosol-dominated regimes in response to varying $PM_{2.5}$ and $NO_x$ concentrations, differentiated by fire-impacted (orange) and non-fire (blue) grid boxes. The dashed line marks the thresholds where half of the grid boxes enter aerosol-dominated regimes. **(b) HOₓ concentrations across different NOₓ concentration bins.** (c) Relationship between the surface $PM_{2.5}$ to $NO_2$ column ratio and the probability of reaching aerosol-dominated regimes, combining both fire-impacted and non-fire grid boxes.

**Reviewer 3**

**General comments:**

Shen et al. simulate O₃ production chemistry during the 2020 California wildfire season using a 3-D chemical transport model (GEOS-Chem) with fire emissions from the Global Fire Emissions Database. The authors find that wildfire smoke aerosols reduce modeled near-surface O₃ production via both heterogeneous uptake of HO₂ and suppression of photolysis rates, and they include these aerosol effects in O₃ photochemical regime assignment based on modeled HOₓ radical termination. Supporting the GEOS-Chem insights with 0-D chemical box modeling of individual plumes, the authors determine that the relative contributions of smoke aerosol effects on O₃ production depend on fire size, plume age, and NOₓ emissions. The authors propose the surface $PM_{2.5}$ to tropospheric $NO_2$ column ratio as an observation-based metric for determining whether O₃ production occurs in an aerosol-dominated regime.

Overall, the manuscript is well written and the topic is within the scope of ACP. The synthesis of four different O₃ photochemical regimes (NOₓ-limited, NOₓ-saturated, aerosol heterogeneous chemistry-inhibited, and aerosol light-limited) into a single framework is a novel approach for understanding wildfire O₃ production chemistry. The analytical methods seem appropriate, although some clarification is required.

We recommend the manuscript for publication pending the consideration of the following points as well as those listed in the Specific Comments section:

- It is not clear what measures were taken to isolate the fire plumes analyzed in GEOS-Chem and F0AM from urban influences on $O_3$ and PM. As the authors state, the interaction between wildfire $O_3$ and PM and urban air quality is not the topic of this paper, and therefore the steps taken to isolate wildfire plume chemistry from urban influences on $O_3$ production should be described explicitly in the methods section.

Thank you for this valuable comment. In this study, we did not explicitly isolate fire plumes from urban influences. Our analysis includes fires occurring across a range of background conditions, e.g., varying $NO_x$ levels. This diversity allows us to examine how interactions among VOCs, $NO_x$ and aerosols affect $O_3$ chemistry.

Most 2020 California wildfires occurred in forest regions, but there are some wildland-urban interface fires such as the SCU and LNU Lightning Complex. Including these fires in the analysis provides additional insights on how background $NO_x$ levels modulate aerosol effects. For example, in Figure 5 (previously Figure 4), we observe light-limited regimes near fires like August complex, Creek and North Complex. In contrast, SCU and LNU Lightning Comple, despite their large size, do not show dominant aerosol effects and are instead classified in $NO_x$-saturated regimes. This phenomenon highlights a competing effect between aerosols and $NO_x$ on $HO_x$ reactions, and helps explain the variability in aerosol impacts across fire events.

We agree that future studies focused on fire–urban plume interactions would benefit from more careful selection of fire plumes to control for urban influences, and we will take this into consideration in future work.

- The manuscript does not address how vertical distributions of smoke and $O_3$ are handled in the models and in the interpretation of the results. Although the analysis focuses only on near-surface $O_3$, the altitude-dependent photochemistry throughout the vertical layers of fire plumes is still relevant. The manuscript would benefit from discussion of this topic; for example, how vertical mixing of lofted $O_3$ contributes to surface enhancements, or whether the fire size categorization considers $PM_{2.5}$ enhancements above the surface (or above the boundary layer) that contribute to aerosol shading effects.

Thank you for your insightful comment regarding the vertical distribution of smoke and $O_3$. In this study, we inject 65% of fire emissions within the boundary layer (Fischer et al., 2014). To assess the sensitivity of our results to this assumption, we conducted additional GEOS-Chem simulations with 100% emissions allocated within the boundary layer. As shown in the figure below, the resulting $O_3$ regimes remain similar to that in Figure 5, with about 4.4%, 19.8%, 60.9% and 14.8% of California falling to $NO_x$-saturated, $NO_x$-limited, heterogeneous chemistry-inhibited and light-limited regimes, respectively.

[Figure]

While based on this sensitivity test, we expect the vertical allocation of emissions has a relatively small impact on the aerosol effects on surface $O_3$, at least on average. Nevertheless, we have revised the sentence to acknowledge the limitation related to fire emission distributions: "We allocate 65% of these fire emissions within the boundary layer (Fischer et al., 2014), **so our findings primarily reflect fires that predominantly impact the boundary layer**." We agree that lofted $O_3$ and $PM_{2.5}$ may influence surface $O_3$ through processes such as vertical mixing and aerosol shading. While such dynamics are beyond the scope of our current study, they present important avenues for future work. Case studies examining the impact of lofted smoke layers on surface $O_3$ would be particularly valuable.

**Specific Comments:**

Line 16: Suggest replacing "positive effects" with "$O_3$ enhancement" and "negative effects" with "$O_3$ suppression" here and throughout the text. The positive/negative terminology is not explicitly defined and thus could be misconstrued as good/bad impacts on air quality.

Thank you for the recommendation. We have revised the sentence as: "While smoke aerosols typically inhibit $O_3$ production through heterogeneous chemical and radiative pathways, we find that for most fires, **the $O_3$ enhancement driven by precursor emissions outweighs these aerosol-driven suppression effects**." We have also updated other positive/negative expressions in the manuscript.

Line 40: Aldehyde photolysis and alkene photolysis are also important pyrogenic $HO_x$ sources (Robinson et al. *ES&T* 2021)

Thank you. We have added the reference and revised the text as follows: "Fires not only emit abundant $O_3$ precursors but also provide important sources of hydrogen oxide radicals ($HO_x$ = OH + $HO_2$ + organic peroxy radical ($RO_2$)) through the photolysis of nitrous acid (HONO), formaldehyde (HCHO), **other aldehydes and $O_3$, as well as the ozonolysis of alkenes** (Jaffe and Wigder, 2012; **Robinson et al., 2021**; Xu et al., 2021). These radicals catalyze the chain oxidation of VOCs in the presence of $NO_x$ to produce $O_3$ (Xu et al., 2021)."

Line 69–70: Reference(s) are needed for the statement that "dense smoke can create a dark environment that makes $O_3$ production limited by light" (possibly from the list of references on Line 51).

We have added the reference: "Moreover, dense smoke can create a dark environment that makes $O_3$ production limited by light (Jiang et al., 2012)."

Line 73: The statement "introducing two new regimes" should be reworded, as it suggests that both the aerosol chemistry-inhibited and light-limited regimes are original to this work. The aerosol-inhibited regime has previously been described in the literature (e.g., Ivatt et al. 2022, as referenced on Lines 67–69).

Thank you for the suggestion. We have revised the statement as follows:

"Therefore, in this study, we refine the **current $O_3$ regime framework by introducing a new regime–the light-limited regime** to better represent the role of aerosols in $O_3$ formation."

Line 99–101. While the authors state specifically that ethene and ethyne chemistry are included in the GEOS-Chem mechanism, they do not mention whether the GEOS-Chem version they used includes furanoid chemistry (Carter et al. 2022). Furanoid compounds are a major product of biomass burning that contribute to $O_3$ and PM formation (Romanias et al. 2024).

Thank you for pointing this out. The version of GEOS-Chem used in our study does not include furanoid emissions and chemistry. We acknowledge that furanoids can contribute to $O_3$ and PM in biomass burning plumes and will consider incorporate this chemistry in future work.

Line 105: The allocation of 65% of fire emissions to the boundary layer is not representative of all fires; the authors should clearly state that their model is specific to fires that mainly impact the boundary layer.

Thank you. We have added this limitation statement as follows: "We allocate 65% of these fire emissions within the boundary layer (Fischer et al., 2014), **so our findings primarily reflect fires that predominantly impact the boundary layer.**"

Line 121–123: Clarification is need on what "iterated over" means in this context.

We have revised this sentence to reduce confusion: "**GEOS-Chem assumes the same $\gamma_{HO_2}$ for all aerosol types**, including organic carbon (OC), BC, sulfate-ammonium-nitrate, sea salt separated in two size bins and mineral dust in seven size bins."

Line 135–138: Further details are necessary in the text to clarify the time period (full year, 1 pm local?) and location (only pixels in which there is an EPA monitor?) over which the reported $O_3$ values are averaged. The reported $O_3$ standard deviations should only have one significant figure. It would be useful to include a scatter plot showing the correlation between model and in-situ $O_3$ in Figure S1, especially since the $R^2$ for this correlation is reported in the text.

The evaluation was performed for 2020 at 1 PM local time, using only grid cells that contain an EPA monitoring site. We have included these details and adjusted the significant figures of the standard deviations to one digit. We corrected the $R^2$ value by calculating is using the Pearson correlation coefficient instead of the Spearman coefficient. The revised text now reads:

"We first evaluate GEOS-Chem predicted $O_3$ with daily ground measurements from the EPA AQS, as presented in Figure S1. The comparison is conducted between AQS sites and the corresponding GEOS-Chem grid cells for the year 2020 around 1 PM local time. The modeled average $O_3$ levels in California for 2020 are approximately $48 \pm 4$ ppb, in good agreement with ground observations of $44 \pm 9$ ppb ($R^2$ of 0.64)."

[Figure]

[Figure]

**Figure S1. (a)** Comparisons of 2020 GEOS-Chem simulations ($\gamma_{HO_2}$ = 0.2) and EPA AQS ground measurements for $O_3$ around local 1 PM. The shaded areas represent GEOS-Chem simulations, and the dots indicate ground measurements. **(b)** Scatter plot of annual mean $O_3$ from GEOS-Chem simulations versus AQS observations.

Line 143–148: The starting altitudes (injection heights) for the HYSPLIT forward trajectories are not stated. The authors should mention if the trajectory heights (i.e., whether the plume remained in the boundary layer) were considered in the F0AM – GEOS-Chem comparison.

Thank you for your comment. The HYSPLIT model was run at an injection height of 1000 m. We have clarified this in the revised manuscript: "**The HYSPLIT model is run at an injection height of 1000 m and initialized** at the same time of the day (18 UTC). In the absence of strong wind variability, the predicted plume trajectories should reasonably represent the progression from the near to far field of fires."

We did not explicitly constrain the trajectory heights in the F0AM – GEOS-Chem comparison as out study does not assume that all fire emissions remain in the boundary layer. In GEOS-Chem, we allocate 65% fire emissions in the boundary layer and 35% in the free troposphere. Moreover, since both the initial species concentrations and aerosol effects on J-values in F0AM are constrained by GEOS-Chem outputs, the influence of aerosol located above the boundary layer on photolysis is inherently included. Therefore, we believe it is not necessary to explicitly account for trajectory heights in the comparison.

Line 152–156: It is unclear whether all 470 fire plumes (described in Section 2.2) were modeled individually in F0AM, or just a subset of these plumes. F0AM version 4.3 includes a mechanism based on MCM v3.3.1 with additional biomass burning chemistry ("MCMv331_AllRxns_NOAABB"). The text should specify whether this additional chemistry has been included; if it has not, the authors should consider the sensitivity of the F0AM model results to the addition of this chemistry. Further details are needed about the aerosol sizes used in F0AM: Does the "monodisperse size distribution" refer to aerosol size *within* the different aerosol type/size categories described in Section 2.1? How was the aerosol radius determined for each aerosol type, and what are these radii? Is the F0AM model sensitive to the choice of aerosol radii?

The F0AM and GEOS-Chem comparison was conducted to validate our F0AM setup and show that when accounting for the chemistry and dilution, F0AM is capable of reproducing GEOS-Chem results. We included a subset of fire plumes for comparison as shown in Fig. S5, ensuring coverage of fires across a range of scales to capture diverse plume behaviors. We did not include the additional "MCMv331_AllRxns_NOAABB", which was developed in Decker et al. (2019) to simulate nighttime chemical transformation in biomass burning plumes. As our study focuses primarily on daytime ozone chemistry around 13:30 local time, we consider the inclusion of this nighttime-specific mechanism unnecessary and outside the scope of our analysis.

Regarding the monodisperse size distribution: Yes, the monodisperse size distribution in F0AM refers to a single representative aerosol radius assumed for each aerosol type, as described in Section 2.1. In the F0AM–GEOS-Chem comparison, we use the effective aerosol radii output from the GEOS-Chem HISTORY diagnostics to evaluate whether F0AM can reproduce GEOS-Chem simulations.

In the F0AM fuel type analysis, we focused primarily on BC and OC from fire emissions. We assigned radii of 0.035 μm and 0.1 μm for BC and OC, respectively. These values closely match the average effective aerosol radii over California in 2020 at 13:30 local time, which are 0.039±0.005 μm for BC and 0.1±0.009 μm for OC. We have added a description of the aerosol radii used in F0AM in Section 2.3: "…Unlike previous setup using GEOS-Chem outputs, here we initiate F0AM with gas phase pollutants and aerosols (primarily OC and BC) for various fire types according to the GFED emission factors. **We adopt aerosol effective radii of 0.035 μm for BC and 0.1 μm for OC, values that closely match GEOS-chem averages over California in 2020 at 1:30 PM local time, and assume a particle density of 1.3 g cm$^{-3}$.**"

To test the sensitivity to aerosol size, we performed additional simulations in which aerosol radii were halved and doubled. The results shown below suggest similar magnitudes of overall fire effects and aerosol effects as those in Fig. 2 of Section 3.2. Furthermore, we observe a consistent trend: boreal forest burning requires the greatest PM enhancement for radiative effect to exceed chemical effect, followed by peat and temperate forest. In contrast, deforested/tropical forest, agricultural waste and savanna tend to exhibit stronger radiative effect even with small PM enhancements from fires. These sensitivity tests indicate that the choice of aerosol radii has a minor impact on F0AM results.

**(a) Simulation with aerosol radii reduced by half**

[Figure]

**(b) Simulation with aerosol radii increased twofold**

[Figure]

Line 163–164: The authors may consider adding justification for scaling J-values in F0AM using the GEOS-Chem J-values for HONO and HCHO but not $NO_2$, since the $NO_2$ photolysis rate is also critical for $O_3$ production.

Thank you for the comment. We use GEOS-Chem J-values in F0AM to account for the effect of aerosols on photolysis rates, since the Hybrid method in F0AM does not include aerosol corrections. Our strategy is to adjust for both UV and visible wavelengths. While $NO_2$ photolysis is indeed important for $O_3$ production, we chose not to scale its J-value directly because the photolysis reactions differ between the mechanisms: GEOS-Chem represents $NO_2$ photolysis as $NO_2 + h\nu \rightarrow NO + O_3$, whereas in MCM (used in F0AM), it is $NO_2 + h\nu \rightarrow NO + O$. Due to this mechanistic inconsistency, directly applying GEOS-Chem $J(NO_2)$ in F0AM could introduce further error. Thus, instead, we use the J-value for HONO, which also photolyzes in the visible range, as well as the J-value for HCHO, which represents UV-affected photolysis. The good agreement between F0AM and GEOS-Chem results (Figure S5) suggests that this scaling approach reasonably captures the aerosol impact on photochemistry.

Line 171–181: In the F0AM fresh plume analysis using GFED emission factor inputs, it is not clear how the aerosol effects were calculated. The authors should also indicate whether the scaling of pollutants/inputs "to achieve aerosol concentrations ranging from 1 to 300 µg m$^{-3}$" is based aerosol concentrations at the time of emission or at a certain plume age.

Thank you for your comment. In the F0AM fresh plume analysis, aerosol effects are calculated using the same approach as in the F0AM–GEOS-Chem comparison and in Table S2. Specifically, we conduct three F0AM simulations: F0AM-base, F0AM-no_rad and F0AM-no_chem simulations. The differences between F0AM-no_rad/F0AM-no_chem and F0AM-base are used to quantify the aerosol radiative and chemical effects, respectively. The scaling of pollutants is based on aerosol concentrations at the time of emission. We have included this information in the Method section. The revised text reads:

"We then scale all pollutants to achieve aerosol concentrations ranging from 1 to 300 µg m$^{-3}$ **at the time of emission**, allowing us to explore how aerosol effects vary with fire intensity. In this approach, we set only the initial chemical and physical parameters and run the model for one hour, focusing specifically on the characteristics of fresh plumes. Photolysis rates, which we cannot directly constrain in scenarios with and without fires, are estimated based on the relationship between photolysis rate reduction and PM$_{2.5}$ mass as derived from GEOS-Chem (Figure S2). To prevent the build-up of secondary species, we set a one-day lifetime for all species by applying a first-order dilution rate of 1/86400 s$^{-1}$ and background concentrations at zero. **Aerosol effects are calculated following the same method as in the F0AM–GEOS-Chem comparison.**"

Line 207: The PM enhancement size bins have not been defined. Section 2.2 (Fire Plume Evolution Analysis) may be a good location to define the PM enhancement size bins, along with clarification as to whether the fire size assignment is based on only the PM enhancement in the pixel nearest to the source for a given fire, or for all individual downwind pixels.

Thank you for your suggestion. We have added more description of the PM enhancement size bins before Line 207 to improve clarity. The new text reads:

"**Fire pixels are categorized based on PM enhancement (ΔPM$_{2.5}$), calculated as the difference in PM$_{2.5}$ mass between the BASE and NO_FIRE simulations for each individual grid cell. Specifically, ΔPM$_{2.5}$ values of <50 µg m$^{-3}$, 50–100 µg m$^{-3}$, 100–200 µg m$^{-3}$ and >200 µg m$^{-3}$ are used to classify small, medium, large and extreme fire pixels, respectively.**"

Line 219–227 (Figure 1): It has not been specified whether GEOS-Chem pixels containing smoke plumes and strong urban influence were identified and excluded before calculating the averages shown in Figure 1. In other words, if the NOx in the model plumes is not exclusively from fire emissions, then the authors should address how NOx from background or urban sources influences the results shown in Figure 1. Additionally, it would be useful to discuss whether there are any trends in $O_3$ observed in plumes of the same fire size category (i.e., plumes that have the same absolute $\Delta PM_{2.5}$) that have different normalized excess mixing ratios of $PM_{2.5}$ ($\Delta PM_{2.5}/\Delta CO$) as a way to probe the impacts of relative rather than absolute aerosol loading.

We appreciate the suggestion. The primary goal of the figure is to capture the average aerosol effects for fires of different sizes and at varying distance from the source. While distinguishing fire pixels with and without strong urban influences could provide additional insight, determining the chemical signature and thresholds for urban plumes involves large uncertainties. Moreover, the newly added fire map (revised Figure 1) demonstrates that most fires indeed occur in forested regions, making it unlikely that urban plumes substantially affect the average aerosol effects we seek to quantify.

Regarding the second point, we have calculated the $\Delta PM_{2.5}/\Delta CO$ within each fire size category and investigated the associations between aerosol effects on $O_3$ and the ratio, as shown in the figure below. Radiative effect is generally minor for small to large fires. The radiative effect for extreme fires and the chemical effect for all fire sizes both initially increase with $\Delta PM_{2.5}/\Delta CO$ and then decrease. The initial increase likely reflects the influence of greater relative aerosol loading and $\Delta PM_{2.5}/\Delta NO_x$, which favor stronger aerosol effects. However, the aerosol effects start to decrease at some point, maybe because as $\Delta PM_{2.5}/\Delta CO$ increases, more $HO_x$ is consumed in VOCs oxidation and secondary organic aerosol formation and thus limiting its availability for $O_3$ production. Since we do not yet have direct evidence to support this hypothesis, we have opted not to include this interpretation in the manuscript. However, we are interested in exploring this further in future work, particularly by incorporating combustion efficiency and fuel type information.

[Figure]

Line 240-241: The statement "$O_3$ suppression in surrounding areas is transported into these regions" needs clarification and justification.

Thank you for pointing this out. Our intention was to emphasize that $O_3$ concentration is not solely a function of production rate, it also depends on transport and mixing. We apologize for the confusion and have revised the statement to improve clarity.

Here is the updated text: "However, a notable difference exists when comparing large and extreme fire pixels: **while their chemical effects on $O_3$ production are similar in the near field (Fig. S4), extreme fires exert a stronger suppression on $O_3$ concentrations (Fig. 2). This discrepancy likely stems from differences in transport and mixing. In the near field of extreme fires, $HO_x$ levels are low due to limited photochemical activity, making $HO_2$ uptake less influential on $O_3$ production. Nevertheless, extreme fires may cause greater suppression of $O_3$ concentrations near the source. As $O_3$ is transported downwind, this initial suppression can lead to a greater reduction in $O_3$ concentrations despite similar local chemical production. Additionally, extreme fires may experience slower mixing with background air, reducing dilution of ozone-suppressed air and further enhancing the decrease in $O_3$ concentrations.**"

Line 244–257: See comments below for Figure S4, which is confusing. The text should clarify whether the comparisons between F0AM and GEOS-Chem are based on averages of modeled plumes or on individual plumes, and if the latter, how plumes were selected for comparison.

Thank you, we have added these details and revised the text to be: "We first use F0AM to conduct similar experiments with GEOS-Chem output for fire plumes of different scales. **We select 12 fire plumes spanning small, medium, large and extreme cases, and comparisons for each individual fire plume are shown in Figure S5.**"

Line 258–265: Due to the scaling of precursor emissions over two orders of magnitude, the OH exposure (chemical age) over 1 hour of physical time is likely quite different between the 1 $\mu g\ m^{-3}$ $\Delta PM$ plume model and the 300 $\mu g\ m^{-3}$ $\Delta PM$ plume model, for example. Running each model to a specific OH exposure time, rather than a specific physical time, would allow a more robust assessment of the dependence of $\Delta O_3$ on $\Delta PM$ in the plumes.

We appreciate this thoughtful suggestion. However, the primary purpose of running F0AM is to validate the near- and far-field aerosol effects observed in GEOS-Chem (Fig. 2). Near- and far-field are defined based on physical plume age, not chemical age. While we agree that OH exposure is a useful metric for characterizing photochemical age and can offer additional insights into aerosol effects during plume aging, incorporating OH exposure into the validation may introduce unnecessary complexity.

Line 269–274: The authors attribute the variation in aerosol effects on $O_3$, for a given PM magnitude, to $NO_x$ concentrations. It is unclear, however, whether the PM to $HO_x$-precursor (e.g., HCHO) emission ratio the same for all fuel types.

The PM to HCHO emission ratios for boreal forest, peat, temperate forest, tropical forest, agricultural waste and savanna are 5.4, 4.3, 4.8, 3.0, 1.5 and 4.1, respectively. This ratio differs across various fuel types and there is no clear association between this ratio and the PM enhancement threshold at which the radiative effect outweighs the chemical effect. Instead, we find that these thresholds are strongly related to the PM to $NO_x$ ratio. Therefore, we identify PM and $NO_x$ as the primary factors controlling the variability in aerosol effects on $O_3$.

Line 276–281 (Figure 2): The caption should remind the reader that the model results are from F0AM and are near-field/1-hour run time. Including the PM/$NO_x$ emission ratio on each panel would facilitate the comparison described in the text (Lines 267–270).

We have added the PM to NO$_x$ emission ratio on each panel as suggested. The figure caption has also been revised to include this information and to clarify that the results are from F0AM simulations representing a one-hour run time.

[Figure]

Figure 3. The impact of aerosol chemical and radiative pathways on O$_3$ concentrations in response to intensified fires, as indicated by increasing PM enhancement, for various fuel types in the GFED emission inventory. **Results are from F0AM with a one-hour run time.** Orange lines denote overall O$_3$ enhancement due to fires, and green and yellow bars denote decreases in O$_3$ concentrations attributable to the aerosol heterogeneous chemical and radiative pathways. **The PM to NO$_x$ emission ratio is annotated for each fuel type.**

Line 281: The statement that "the dependence of aerosol effects on NO$_x$ is also seen in GEOS-Chem" has not been supported by evidence at this point in the text.

Thank you for your comment. We have added a figure in the supplementary to support this and expanded the description in the main text to provide additional context.

"The dependence of aerosol effects on NO$_x$ is also evident in GEOS-Chem. **Fig. S7 suggests that the radiative effect tends to surpass the chemical effect at high NO$_x$ levels.**"

[Figure]

**Figure S7. Dependence of aerosol chemical and radiative effects on NOₓ concentrations at 20:30 UTC based on GEOS-Chem simulations. The analysis includes all fire pixels within identified plumes.**

Line 287–288: Suggest replacing "has a positive impact on" with "enhances" because the positive/negative terminology is especially vague in reference to Figure 2, where absolute values of $O_3$ enhancement are used.

Great suggestion, fixed.

Line 376–378: The sensitivity test with an $HO_2$ uptake coefficient of 0.1 is a valuable addition to the analysis. However, this uptake coefficient value is still higher than many of the values in Table S1 as well as other studies in the literature (e.g., Tan et al. 2020). It would be useful include a sensitivity test with an uptake coefficient lower than 0.1, as well as the extension of the uptake coefficient sensitivity tests to other components of the analysis (e.g., Section 3.1).

Thank you. We have conducted additional GEOS-Chem simulations using $\gamma_{HO_2}$ of 0.02 to indicate a conservative lower bound. We have also extended the sensitivity analysis for $\gamma_{HO_2}$ of 0.1 and for the reduced fire emissions scenarios (in response to the following comment) to include not only regime calculations but also aerosol effects. Section 3.4 has been renamed as "**3.4 Uncertainties in GEOS-Chem resolved aerosol effects and O₃ regimes**," and the revised text now reads:

"... **Under the $\gamma_{HO_2}$= 0.1 scenario, aerosol effects across fire sizes are similar to Fig. 2: the aerosol chemical effect outweighs the radiative effect for small to large fire pixels, while extreme fire pixels show a pronounced radiative effect (Fig. 11 (a) (b)). Although the overall fire effect reduces O₃ net production rate, its influence on O₃ concentrations is minimal.** The spatial pattern of photochemical regimes remains largely unchanged under this scenario (Fig. S12 (a)).

**Given that $\gamma_{HO_2}$ measured for single-component organics likely underestimates values for ambient aerosols, the $\gamma_{HO_2}$ = 0.02 case is tested as a conservative lower bond. Under this assumption, aerosol chemical and radiative effects on O₃ concentrations become comparable for most fire pixels, whereas extreme fire pixels continue to exhibit a pronounced radiative effect (Fig. 11 (c) (d)). Although this strong radiative effect suppresses O₃ production in near-field extreme fire pixels, O₃**

**concentrations still increase, possibly due to transport of ozone produced earlier near the fire source. With this substantially reduced uptake coefficient, the spatial extent of heterogeneous chemistry-inhibited regimes decreases markedly. Nevertheless, overall aerosol influences remain important, with 31% of California falling into aerosol-dominated regimes (Fig. S12 (b)).** Future research measuring $\gamma_{HO_2}$ for smoke aerosols is needed to better constrain this parameter.

Furthermore, we evaluate GEOS-Chem simulations of $PM_{2.5}$ with ground-based measurements from EPA's AQS. ... **Specifically, GFED fire emissions are adjusted by dividing total emissions by 0.6, 1.2, 4.1, 2.4, 1.4 and 0.9 for July through December, respectively. Despite the substantial reduction in overall fire emissions, the aerosol and total fire effects on $O_3$ concentrations for most fires remain consistent with Fig. 2, whereas the radiative effect for extreme fires declines markedly in both the near and far field due to reduced aerosol loading (Fig. S11 (e) (f)).** Aerosol-dominated regimes still accounted for about 7%, 54% and 17% of the total area in August, September and October, respectively (Fig. S13). Notably, aerosol-dominated regimes remain dominant in September during the 2020 fire season."

[Figure]

[Figure]

**Figure S11**. Sensitivity tests of total fire effects and aerosol effects on $O_3$ concentrations **(a, c, e)** and $O_3$ net production rate **(b, d, f)** at 20:30 UTC for fire pixels of different sizes in both near and far field. Panels **(a) and (b)** use $\gamma_{HO_2}$ of 0.1 and include all 470 plumes from September 2020. Panels **(c) and (d)** use a lower $\gamma_{HO_2}$ value of 0.02, based on all September plumes. Panels **(e) and (f)** are based on adjusted GFED emissions and include 1347 plumes identified between July and December 2020.

(a)                                                          (b)

[Figure]

**Figure S12.** Monthly-mean $O_3$ photochemical regimes at 20:30 UTC over California for September 2020, under the BASE scenario with **(a)** $\gamma_{HO_2} = 0.1$ and **(b)** $\gamma_{HO_2} = 0.02$.

Line 380–391: This section addresses some of the uncertainty in the regime calculations by applying a correction to GEOS-Chem PM so that it matches ground site PM observations. It may be worthwhile to test the sensitivity of the average fire effects on $O_3$ (Section 3.1) to this same GEOS-Chem PM correction.

Addressed, please see our response above.

Line 413–415: A comment on the variation, or lack thereof, in $HO_x$ or $HO_x$ precursors across the different $NO_x$ bins would add useful context to Figure 6a.

Thank you for the very helpful suggestion. We have plotted $HO_x$ concentrations across different $NO_x$ bins and discussed the observed variations as follows:

"These results support our earlier findings that in scenarios with high $NO_x$ concentrations, more PM is needed to attain a comparable level of aerosol contribution as observed in low $NO_x$ scenarios. **As $NO_x$ concentrations increase, $HO_x$ levels tend to decrease (Fig. 7 (b)), which necessitates higher PM levels for aerosol effects to surpass the emission effects.**"

[Figure]

Figure 7. (a) Probability of achieving aerosol-dominated regimes in response to varying $PM_{2.5}$ and $NO_x$ concentrations, differentiated by fire-impacted (orange) and non-fire (blue) grid boxes. The dashed line marks the thresholds where half of the grid boxes enter aerosol-dominated regimes. **(b) $HO_x$ concentrations across different $NO_x$ concentration bins.** (c) Relationship between the surface $PM_{2.5}$ to $NO_2$ column ratio and the probability of reaching aerosol-dominated regimes, combining both fire-impacted and non-fire grid boxes.

Line 431–445: Use of the same map projection and/or grids for Figures 4 and 7 would greatly aid the visual comparison described in this section. The combination of a surface measurement (PM) with a tropospheric column measurement ($NO_2$) in the $PM_{2.5}/NO_2$ ratio used to identify regimes in Figure 7 may contribute to some of the discrepancy in regime assignment between Figures 4 and 7. The authors should provide justification for using this combined surface-$PM_{2.5}$/column-NO2 ratio rather than a purely surface-based (ground site $PM_{2.5}/NO_x$) or satellite-based (tropospheric column AOD/$NO_2$) metric.

We have updated the map projection in the original Figure 7 to align with the earlier regime plots. In addition, we corrected a plotting issue in the earlier regime maps and have revised all affected figures in both the main text and the supplementary material.

We also provided a justification for using the combined surface-$PM_{2.5}$/column-$NO_2$ ratio prior to introducing this metric:

"Since $PM_{2.5}$ and $NO_2$ can be derived from ground-based or satellite observations, we explore how their ratio can be used to imply aerosol-dominated regimes. **While a surface $PM_{2.5}/NO_2$ ratio may seem more straightforward based on our analysis, the limited spatial coverage of surface $NO_2$**

**measurements poses a challenge. Tropospheric NO₂ column data, which are closely related to surface sources and have been widely used in O₃ sensitivity analyses (Martin et al., 2004), offer a practical alternative. When combined with high resolution and gapless surface PM₂.₅ estimates derived from the integration of observations and machine learning, the PM₂.₅/NO₂ column ratio serves as a proxy to constrain aerosol effects on near-surface O₃ production. Indeed, we find a clear relationship between this ratio and the likelihood of aerosol-dominated regimes (Figure 7 (c)).** When the ratio (PM₂.₅/NO₂ column) reaches about 20 ($\mu g\ m^{-3}$)/($10^{15}$ molecules $cm^{-2}$), the aerosol-dominated regimes are likely to prevail, and will consistently be dominant at higher ratios."

[Figure]

**Figure 8. Monthly mean O₃ photochemical regimes identified using the surface PM₂.₅ to TROPOMI NO₂ column ratio over California from August to October. Red colors represent aerosol-dominated regimes, while blue colors indicate NOₓ-limited or NOₓ-saturated regimes. Monthly mean PM₂.₅ and NO₂ are used to calculate the ratio, with a threshold of 20 ($\mu g\ m^{-3}$)/($10^{15}$ molecules $cm^{-2}$) applied to identify aerosol-dominated regimes.**

[Figure]

**Figure 5. Monthly-mean GEOS-Chem derived O₃ photochemical regimes at 20:30 UTC (corresponding to 13:30 local time during daylight saving and 12:30 otherwise) over California during the fire season (August to October), when fires are accounted.**

Figure S4: For the 3 panels for a given fire size, there is no description of the difference between the panels (e.g., for the small fire panels, what is the difference between a, b, and c?). Calling the no-fire model results "base" is confusing because in Section 2.1 and Table S2 "base" refers to the *with*-fire model. Arrange the panels in 4 rows (each corresponding to fire size) by 3 columns, and increase the size of each panel.

Thank you for the helpful suggestions. We have clarified the differences between the (a), (b) and (c) panels by noting that they represent separate examples within each fire size category. Fires are categorized based on $\Delta PM_{2.5}$ near fire center: small ($<50$ µg m$^{-3}$), medium (50–100 µg m$^{-3}$), large (100–200 µg m$^{-3}$) and extreme ($>200$ µg m$^{-3}$). We have also clarified the use of "base" to mean the with-fire simulation, consistent with terminology used in Section 2.1 and Table S2. The figure has been rearranged into four rows (by fire size) and three columns (by example), and panel sizes have been increased for readability. The updated figure and caption are as follows:

[Figure]

**Figure S5.** Comparison of aerosol effects on $O_3$ concentrations resolved in GEOS-Chem and in F0AM for fire plumes of different scales. **$PM_{2.5}$ enhancement at plume age of one hour is used to approximately categorize fire sizes: <50 µg m$^{-3}$, 50–100 µg m$^{-3}$, 100–200 µg m$^{-3}$ and >200 µg m$^{-3}$ for small, medium, large and extreme fires, respectively. We include three plumes under each fire size category for comparison, each labeled by the date of the plume.** $O_3$ simulations in F0AM are initiated with chemical concentrations, meteorological parameters and dilution factors derived from GEOS-Chem. **The blue lines indicate $O_3$ concentrations from the base (with-fire) simulation**, while the red, green and yellow lines represent changes in $O_3$ concentrations attributable to total fire effect, chemical effect and radiative effect, respectively. Solid lines represent GEOS-Chem simulations, and dashed lines denote F0AM outputs. To streamline the comparison between GEOS-Chem and F0AM amidst plume dispersions, we focus on the most probable trajectory, selecting a single GEOS-Chem grid box with the highest particle concentrations for each plume age.

**Technical Corrections:**

Line 59: Unnecessary comma after "VOCs"

Removed, thank you.

Line 61: Unnecessary comma after "emissions"

Removed, thank you.

Line 66: Need another dash between "areas" and "often" if using a dash between "loadings" and "typical"

Fixed, thank you.

Line 68: Missing article between "to" and "strong"

We have added "a" between the two words.

Line 115–117: Variable A is not defined. Italicize variable a. Avoid starting sentences with variables.

We have added an explanation of variable A and revised the sentence as follows: "**The first-order rate constant $k$ for the chemical loss of the gas (i.e., $HO_2$) is calculated based on the mean molecular speed ($v$), gas-phase molecular diffusion coefficient ($D_g$), aerosol radius (a), reaction probability upon impacting the aerosol surface ($\gamma$) and aerosol surface area per unit volume of air (A).**"

Line 192: Replace "=" with "-->" in the $HNO_3$ reaction.

Replaced, thank you.

Line 289: Rephrase "concentrations of $NO_2$ column decay" as "$NO_2$ columns decay" for consistency with the units in Figure 3.

Corrected, thank you.

Line 325: Replace dash with comma for consistency with comma on Line 326 (after "light-limited regime")

Replaced, thank you.

Line 434: "Calcification" seems to be a misspelling of "calculation"

Thank you for pointing this out. "Calcification" was a typo—we intended to write "classification" and have corrected it accordingly.

---

## Author Response (AR2)

We thank the reviewers for their constructive feedback that has helped us to improve the manuscript. Our responses to their comments are provided below, with reviewer comments in black, our responses in blue.

Reviewer 1

The authors have satisfactorily addressed most of our comments, adding clarity and detail to the manuscript. Two final, minor comments are listed below:

**1. It is suggested that the methods section explicitly states that fire plumes were not isolated from urban influences in this analysis, in order to probe a range of background NOx levels (as the authors explained in the Response to Reviewers).**

**Reply: We explicitly state this in the revised manuscript:**

We did not explicitly isolate fire plumes from urban influence in order to examine aerosol effects across a range of background $NO_x$ levels.

**2. It is recommended that the manuscript includes a caveat about the absence of furanoid chemistry in the models. This statement would acknowledge that although they are not included in the GEOS-Chem and F0AM mechanisms, furanoid compounds are known to have an important influence on biomass burning plume chemistry (both at nighttime, as described in the Decker et al. paper that the authors correctly referenced in the Response to Reviewers, and in the daytime).**

**Reply: We explicitly state this in the revised manuscript:**

It should be noted that although furanoid compounds markedly influence biomass burning plume chemistry under both daytime and nighttime conditions (Decker et al., 2019; Xu et al., 2021), their reactions are not represented in either the GEOS-Chem version or the MCM mechanism used in this study.